# Maternal thyroid hormone receptor β activation in mice sparks brown fat thermogenesis in the offspring

Rebecca Oelkrug[1], Lisbeth Harder [1], Mehdi Pedaran[1], Anne Hoffmann [2], Beke Kolms[1], Julica Inderhees [3], Sogol Gachkar[1], Julia Resch[1], Kornelia Johann [1], Olaf Jöhren [3], Kerstin Krause [4] & Jens Mittag [1] ✉

It is well established that maternal thyroid hormones play an important role for the developing fetus; however, the consequences of maternal hyperthyroidism for the offspring remain poorly understood. Here we show in mice that maternal 3,3′,5-triiodothyronine (T3) treatment during pregnancy leads to improved glucose tolerance in the adult male offspring and hyperactivity of brown adipose tissue (BAT) thermogenesis in both sexes starting early after birth. The activated BAT provides advantages upon cold exposure, reducing the strain on other thermogenic organs like muscle. This maternal BAT programming requires intact maternal thyroid hormone receptor β (TRβ) signaling, as offspring of mothers lacking this receptor display the opposite phenotype. On the molecular level, we identify distinct T3 induced alterations in maternal serum metabolites, including choline, a key metabolite for healthy pregnancy. Taken together, our results connect maternal TRβ activation to the fetal programming of a thermoregulatory phenotype in the offspring.

Thyroid hormone plays an important role for the development and maintenance of almost all tissues, especially the brain[1–3]. Most remarkably, the thyroid hormone receptors TRα and TRβ are detected in the fetal brain before the embryonal thyroid gland starts producing hormone; consequently, during this developmental window, thyroid hormone needs to be provided by the mother[4]. If this supply is insufficient, e.g., in the case of maternal hypothyroxinemia, the fetal brain can be severely and permanently affected, with bipolar disorders, autism spectrum disorders, attention deficit hyperactivity disorders or epilepsy as possible consequences—often in a sex specific fashion[5–9]. Likewise, increased thyroid hormone levels due to maternal hyperthyroidism or L-thyroxine overtreatment in pregnancy can have negative consequences for the developing fetal brain, resulting for instance in lower child IQ and abnormal brain morphology[5,10,11]. In addition, maternal hypo- and hyperthyroidism can interfere with the

correct setpoint of the offspring's hypothalamus–pituitary–thyroid axis[12]. Despite the accumulating evidence that maternal thyroid dysfunction can negatively impact the health of the developing child, it is still controversially discussed, whether pregnant women should be routinely screened for thyroid dysfunction at the onset of pregnancy[13]. Consequently, additional preclinical and clinical studies are urgently required to define the precise phenotypical consequences of maternal thyroid disorders, as they can identify previously not recognized outcomes of these conditions, e.g. elevated blood pressure in the offspring[14,15].

Body temperature regulation is a major target of thyroid hormone action in the adult animal[16]. The hormone affects heat dissipation[17,18] as well as thermogenesis, where it acts both in muscle as well as brown adipose tissue (BAT), a tissue specialized for heat production to maintain body temperature[19–23]. In particular, the hormone's action on

[1]Institute for Endocrinology & Diabetes – Molecular Endocrinology, Center of Brain Behavior and Metabolism (CBBM), University of Lübeck, Ratzeburger Allee 160, 23562 Lübeck, Germany. [2]Helmholtz Institute for Metabolic, Obesity and Vascular Research (HI-MAG) of the Helmholtz Zentrum München at the University of Leipzig and University Hospital Leipzig, Philipp-Rosenthal-Straße 27, 04103 Leipzig, Germany. [3]Bioanalytic Core Facility - Center of Brain Behavior and Metabolism (CBBM), University of Lübeck, Ratzeburger Allee 160, 23562 Lübeck, Germany. [4]Department of Endocrinology, Nephrology, Rheumatology, University of Leipzig Medical Center, 04103 Leipzig, Germany. ✉e-mail: jens.mittag@uni-luebeck.de

BAT thermogenesis has been of interest in the light of the obesity pandemic[19,24], as the tissue is potentially capable of converting stored fatty acids to heat primarily through uncoupling protein 1 (UCP1), which causes mitochondrial uncoupling of the respiratory chain. While BAT is found in rodents and human newborns[19,25], its presence in adult humans is highly variable and the contribution to whole body energy expenditure questionable[26–28]. However, why some individuals seem to possess activatable amounts of BAT and others do not remains enigmatic, but fetal events have emerged as possible regulators in rodent studies[29].

Here we demonstrate that elevated levels of maternal thyroid hormone enhance the offspring's physiological mechanisms to maintain body temperature, including overactive BAT thermogenesis, which is already detectable early after birth. This fetal programming mechanism depends on maternal thyroid hormone receptor β (TRβ) signaling as the opposite effect was observed in offspring of mothers lacking this receptor.

## Results

### Effects of maternal T3 on dams and pups

To characterize the effects of maternal hyperthyroidism on offspring development and phenotype, we treated pregnant dams from conception to gestational day (GD) 17 with 0.5 mg/L T3 in drinking water (Fig. 1a). This resulted in elevated serum T3 levels, lower serum T4 and completely suppressed hypophyseal thyroid-stimulating hormone β (*Tshb*) mRNA expression in treated dams (Fig. 1b), indicative of hyperthyroidism, which was confirmed by elevated gene expression of hepatic deiodinase type I (*Dio1*) and thyroid hormone inducible hepatic protein *(Thrsp)* as well as increased liver and heart weight (Supplementary Fig. 1a, b). The T3-treated mothers gained more weight during pregnancy, accompanied by increased food and water intake (Fig. 1c) and improved glucose clearance at GD14-15 (Fig. 1d). These metabolic alterations in the T3-treated dams affected embryonal weight as well as embryonal hepatic gene expression at embryonal day (E) 17.5 in both sexes (Fig. 1e and Supplementary Fig. 1c), whereas male to female ratio and litter size were not affected (Supplementary Fig. 1d and Fig. 1f). However, the T3-treated dams gave birth one day later as compared to controls (GD20 vs. GD19), which might have contributed to increased birth weight in their offspring (Fig. 1g). To test whether gestational T3 treatment affected postnatal maternal care, we conducted a pup retrieval test, which did not reveal any impairments (Fig. 1h). Most remarkably, at postnatal day (P5), the offspring of T3-treated dams ("mT3 offspring") had an increased interscapular BAT (iBAT) temperature two minutes after taken from the nest in both sexes (Fig. 1i, j), while the initial temperature after removal was similar (control males: 34.62 ± 0.12 °C, mT3 males: 34.40 ± 0.31, control females: 34.28 ± 0.19 and mT3 females: 34.86 ± 0.21 °C).

### Postnatal phenotype of offspring of T3 treated mothers

Despite the initial weight difference at birth, we did not observe any effect on body weight development or body composition in either sex (Fig. 2a, b). Somewhat unexpected from previous studies[30], we did not find any effect of maternal T3 on serum T4 and T3 as well as pituitary *Tshb* mRNA expression in offspring of either sex (Fig. 2c), and expression of TH sensitive hepatic genes was also not altered (Supplementary Fig. 1e). Interestingly, while the iBAT hyperactivity was transiently not observed at 4 weeks of age due to an increased iBAT thermogenesis in the controls, presumably induced by the cold stress caused by weaning (Supplementary Fig. 1f), it returned in adult animals of both sexes at 10 weeks of age (Fig. 2e, f), with no effect on rectal body temperature (Fig. 2g). This was corroborated by normal oxygen consumption, respiratory quotient, and average daily energy expenditure in both sexes (Fig. 3a–d and Supplementary Fig. 1g, h), suggesting that the iBAT hyperactivity has no net effect on whole body

energy expenditure. Furthermore, resting metabolic rate at 30 °C and food intake were also not changed (Fig. 3e, f). Interestingly, we observed a reduced basal serum glucose and faster glucose clearance exclusively in the male mT3 offspring (Fig. 3g, h) with a similar response to insulin albeit on a lower baseline (Fig. 3i), while female offspring were unaffected (Supplementary Fig. 1i). The altered glucose metabolism was not caused by browning of inguinal white adipose tissue, as we could not detect any UCP1 protein in this tissue and *Ucp1* mRNA levels were at background levels in either condition. Taken together these data demonstrate that maternal T3 levels positively program iBAT thermogenesis in both sexes, while improving glucose metabolism exclusively in the male offspring.

### Role of maternal thyroid hormone receptor β

To test for a possible role of TRβ in the maternal or the offspring compartment, we used TRβ-/- females as dams and compared the offspring to those of TRβ+/- females. Both females were mated with TRβ+/- males, allowing us to analyze the resulting TRβ+/- (hereafter referred to as "controls", as they are indistinguishable from TRβ+/+ wildtypes) and TRβ-/- offspring (Fig. 4a). While total serum T3 and T4 were surprisingly normal in the pregnant TRβ-/- dams as compared to TRβ+/- females, pituitary *Tshb* and *Dio2* mRNA showed the expected upregulation due to the lack of TRβ mediated negative feedback loop (Fig. 4b), a hallmark of thyroid hormone resistance (RTH)[31]. Litter size and male to female ratio were not affected by maternal genotype (Supplementary Fig. 2a).

For the subsequent analysis, we focused solely on the male offspring, as we were most interested in the fetal programming of iBAT and glucose sensitivity, and the latter was not affected in the female mT3 offspring. Our data revealed that body weight was lower in adult male TRβ+/- mice from TRβ-/- dams as compared to the males from control dams (Fig. 4c left); however, there was no difference in TRβ-/- male offspring born by control or TRβ-/- dams (Fig. 4c right), although the males lacking TRβ were generally lighter. Body composition analysis by NMR showed significantly increased fat mass in TRβ-/- mice, irrespectively of the mother's genotype; the post hoc tests however were not significant (Fig. 4d, Supplementary Data 1). As maternal as well as offspring TRβ deficiency are associated with alterations in the hypothalamus-pituitary-thyroid axis[30,32], we next analyzed serum total T4 and total T3 concentrations. The results showed the expected increase in total serum T3, T4 and pituitary *Tshb* (Fig. 4e) as well as expected repression of hepatic *Dio1* (Supplementary Fig. 2c) in mice lacking TRβ; however, maternal TRβ deficiency had no effect on these parameters.

Most remarkably, when we tested the iBAT thermogenesis, we observed strongly reduced surface temperatures in all offspring of TRβ-/- mothers irrespectively of their own genotype (Fig. 4f, g), clearly demonstrating that the fetal programming effect is mediated by maternal TRβ and independent of fetal TRβ. There was no consequential effect of the altered iBAT thermogenesis on body or tail temperature (Fig. 4h and Supplementary Fig. 2d), oxygen consumption, respiratory quotient, daily energy expenditure, food intake, or resting metabolic rate at 30 °C (Fig. 4i, j and Supplementary Fig. 2e). Interestingly, the lack of maternal TRβ had no effect on glucose tolerance in the offspring (Fig. 4k), indicating that the iBAT and the glucose sensitivity phenotype are not causally related.

### Phenotyping of brown adipose tissue function

To identify a molecular reason for the altered iBAT function, we performed microarray gene expression analysis in the iBAT of all male offspring. The volcano plot analysis showed several genes that were differentially expressed between control or mT3 offspring (Fig. 5a, not corrected for multiple testing) and offspring of control or TRβ-/- dams (Fig. 5b, not corrected for multiple testing), respectively. However, when the analyses were combined to identify iBAT genes that

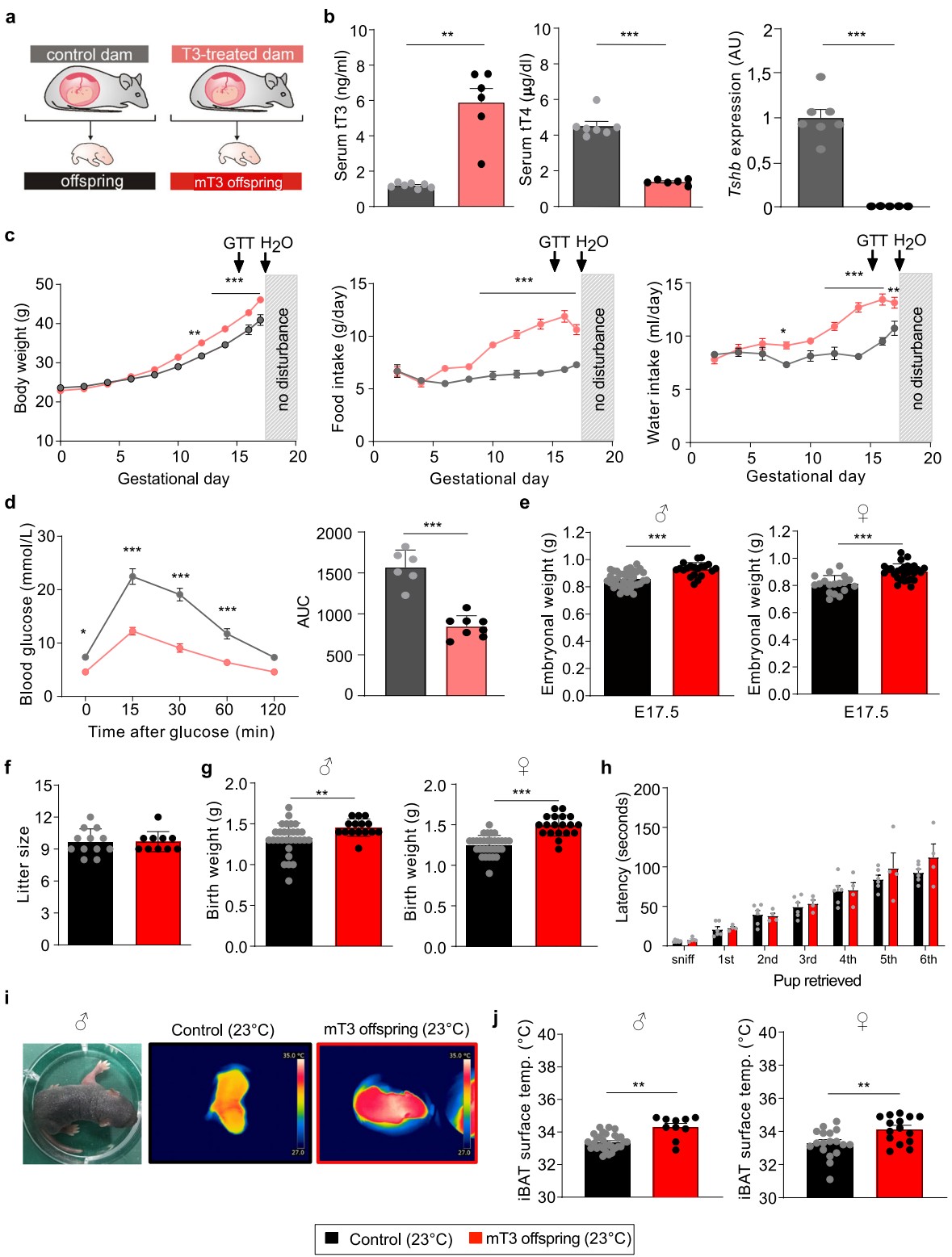

correlated with the observed phenotype, i.e. regulated in opposite directions in mT3 offspring versus TRβ-/- offspring, only 19 genes remained (Fig. 5c, Supplementary Table 1). When those were corrected for multiple testing, only Olfr237-ps1, an olfactory receptor pseudo-gene remained, which is unlikely to explain the observed effect. Moreover, neither genes governing glucose metabolism (GO pathway: 0006006) nor adrenergic signaling (GO pathway: 0071875) were consistently regulated with the phenotype (Supplementary Fig. 3a, b).

Given the lack of an endogenous iBAT gene expression defect that could explain the phenotype, we performed an in-depth character-ization of the tissue. Our findings showed no difference in iBAT weight (Supplementary Fig. 3c, d), no difference in protein expression of UCP1 (Fig. 5d) or oxidative chain complexes (Supplementary Fig. 3e, f) in either maternal condition. When we then analyzed iBAT cAMP levels as a measure of sympathetic stimulation, we found normal levels in mT3 offspring of both sexes, but reduced levels in offspring of TRβ-/-

**Fig. 1 | Effects of maternal T3 on dams and pups. a** Breeding scheme of the T3-treated and control dams. **b** Serum levels of total T3 and T4 as well as pituitary *Tshb* mRNA expression in pregnant T3-treated (light red, *n* = 6 for T3/T4 and *n* = 5 for *Tshb*) and control (gray, *n* = 7) dams at gestational day 17 (GD17). **c** Body weight development, food and water intake during pregnancy until GD17 with the timing of the glucose tolerance test (GTT) and the end of the T3 treatment indicated by H$_2$O (*n* = 10 per group). **d** Glucose i.p. tolerance test at GD14–15 (left panel) with area under the curve (AUC, right panel) in the pregnant dams (*n* = 6 (control) and 8 (T3-treated dams)). **e** Weight of the male and female embryos from control (black, *n* = 36 males and *n* = 18 females) or T3-treated mothers (red, *n* = 20 males and *n* = 28 females) at embryonic day E17.5. **f** Litter size (*n* = 12 (control) and 10 (T3-treated dams)) and **g** birth weight (*n* = 29 control males and 31 control females, *n* = 15 males

of T3-treated dams and *n* = 19 females of T3-treated dams) in both groups. **h** Pup retrieval test indicating the time after removing the litter from the nest to first sniff or retrieval of the respective pup, showing same maternal care in both mothers (*n* = 6 control dams and *n* = 4 T3-treated dams). **i** At postnatal day 5 (P5) the offspring was removed from the nest for 2 min and studied by infrared thermography with **j** quantification of iBAT surface temperature (n = 30 control males and 18 control females, *n* = 10 males of T3-treated dams and *n* = 15 females of T3-treated dams). All values are mean ± SEM with individual values in circles. *p < 0.05; **p < 0.01 and ***p < 0.001 for maternal T3 treatment. Biological replicates were obtained from 3 to 4 litters per group. AU arbitrary units. Statistical details are provided in Supplementary Data 1. Source data are provided as a Source data file.

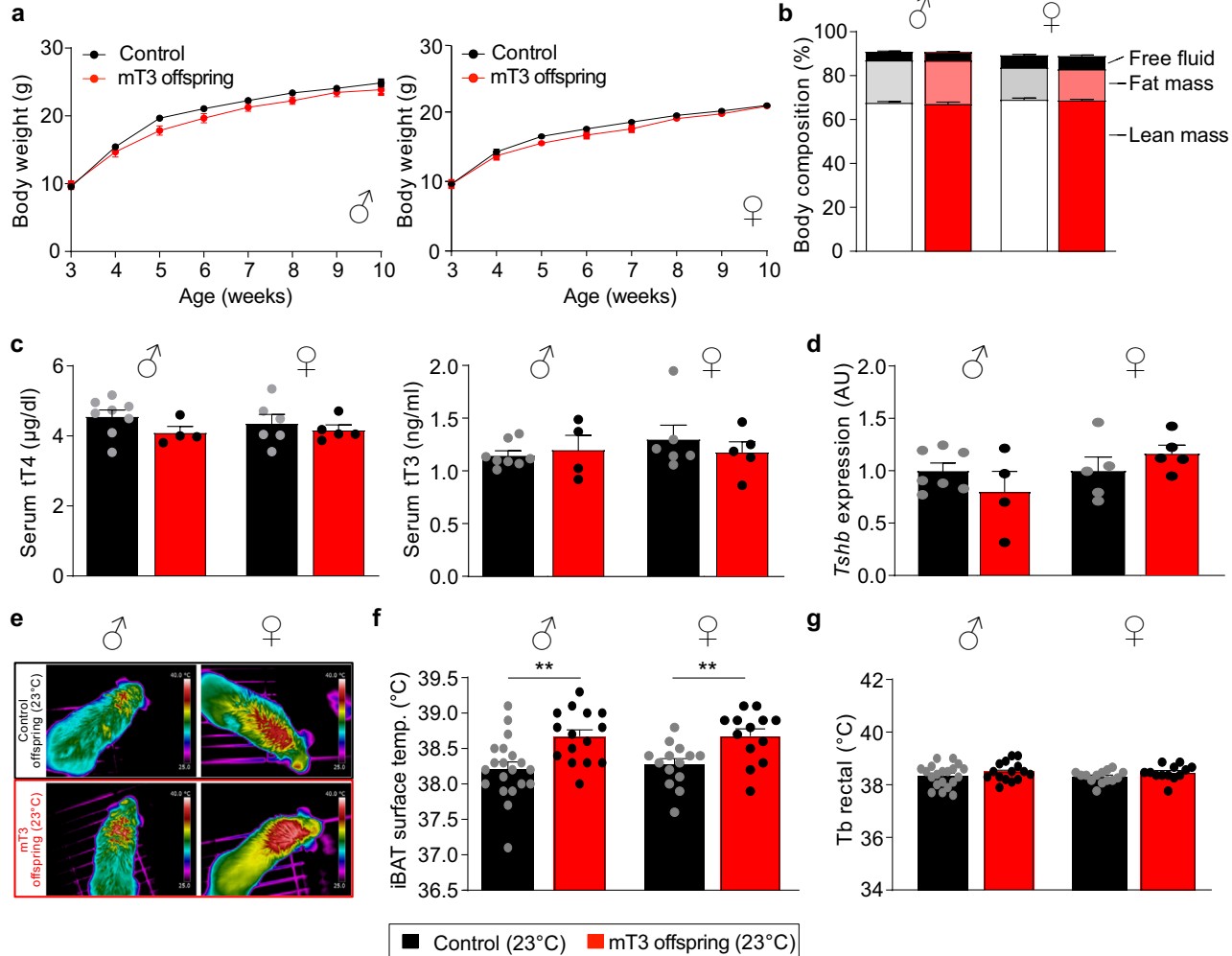

**Fig. 2 | Postnatal development of offspring of T3-treated mothers. a** Growth curve of males and females born by control (black, *n* = 25 males and *n* = 21 females) or T3-treated mothers (red, *n* = 15 males and *n* = 14 females). **b** Body composition of those offspring (black controls: *n* = 12 males and *n* = 8 females; red mT3 offspring: *n* = 11 males and *n* = 8 females per group) and **c** serum total T3 and total T4 concentrations (controls: *n* = 8 males and *n* = 6 females, mT3 offspring: *n* = 4 males and *n* = 5 females) as well as **d** pituitary *Tshb* mRNA expression of these offspring at the age of 5 months (controls: *n* = 7 males and *n* = 5 females, mT3 offspring: *n* = 4 males and *n* = 5 females). **e** Infrared thermography of the animals at >10 weeks of age.

**f** Quantification of the BAT infrared thermography (control: *n* = 20 males and *n* = 14 females, mT3 offspring: *n* = 15 males and *n* = 13 females). **g** Rectal body temperature (Tb) in these animals (control: *n* = 20 males and *n* = 14 females, mT3 offspring: *n* = 15 males and *n* = 13 females). All values are mean ± SEM with individual values in circles. **p < 0.01 for maternal T3 treatment. mT3: offspring of maternally T3-treated dams. Biological replicates were obtained from 3 to 4 litters per group. AU arbitrary units. Statistical details are provided in Supplementary Data 1. Source data are provided as a Source data file.

mothers irrespective of the own genotype (Fig. 5e). This demonstrates that the iBAT hypoactive phenotype in offspring of TRβ-/- mothers is caused by a reduced sympathetic stimulation, and might thus be mechanistically different from the effect observed in offspring of T3-treated mothers.

We thus continued to test the iBAT of the mT3 offspring in detail. A norepinephrine (NE) stimulation test of these animals confirmed that the maximal BAT response and thus the level of recruitment is not altered in these animals (Fig. 5f) as expected from the microarray and Western Blot data. Most interestingly, in

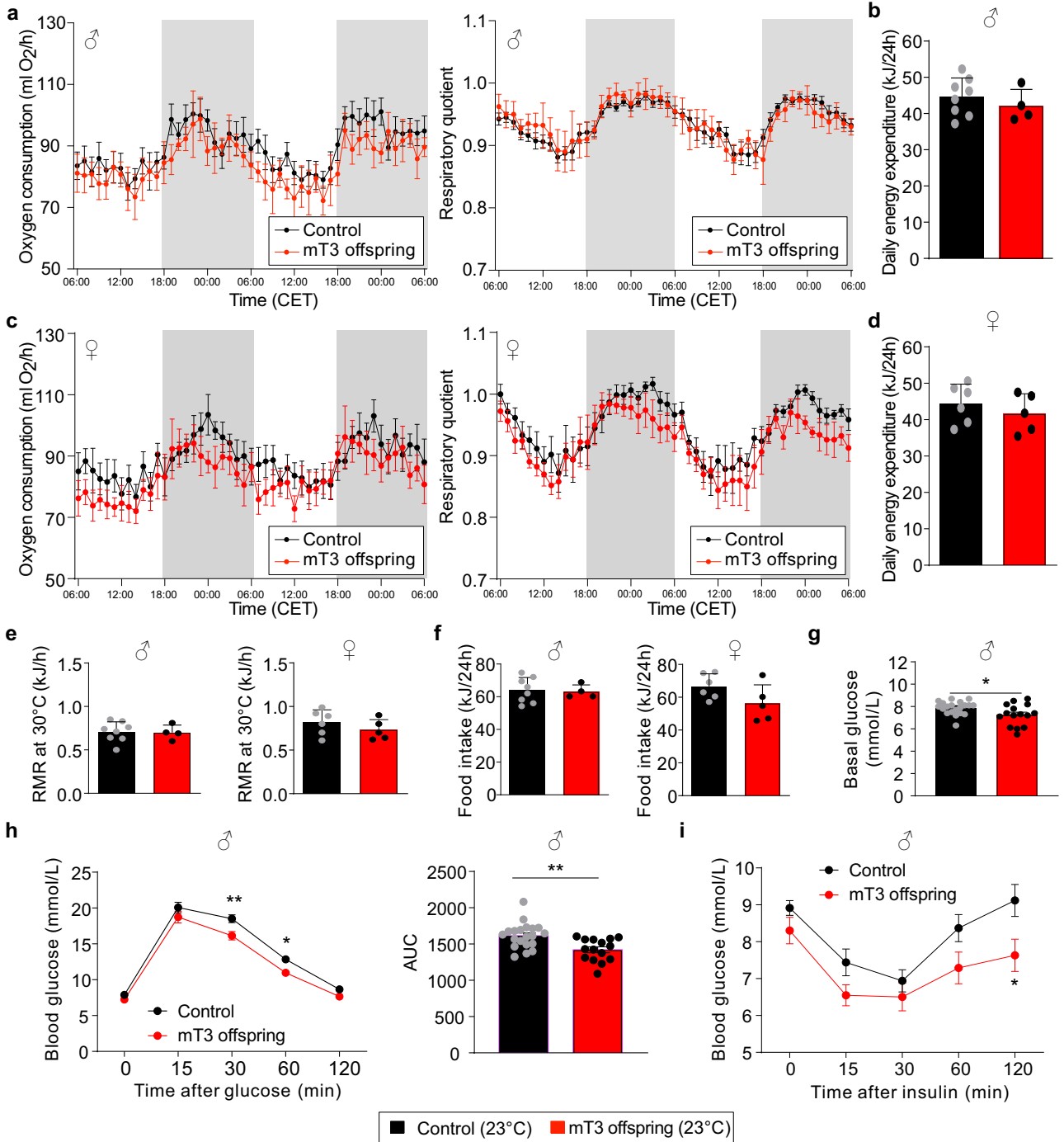

**Fig. 3 | Metabolic phenotype of offspring of T3-treated mothers. a** Oxygen consumption (left) and respiratory quotient (right) of the male offspring born by control (black) or T3-treated mothers (red) over the course of 2 consecutive days at the age of 4–5 months. Gray shadows indicate night-time activity period ($n = 8$ (control) and 4 (mT3 offspring)). **b** Daily energy expenditure per 24 h in these animals (n = 8 (control) and 4 (mT3 offspring)). **c** Oxygen consumption (left) and respiratory quotient (right) of the female offspring over the course of 2 consecutive days at the age of 4–5 months. Gray shadows indicate night-time activity period ($n = 6$ (control) and 5 (mT3 offspring)). **d** Daily energy expenditure per 24 h ($n = 6$ (control) and 5 (mT3 offspring)). **e** Resting metabolic rate measured at thermoneutrality (30 °C) in male and female offspring (males: $n = 8$ (control) and 4 (mT3 offspring); females: $n = 6$ (control) and 5 (mT3 offspring)). **f** Food intake (males: $n = 8$ (control) and 4 (mT3 offspring); females: $n = 6$ (control) and 5 (mT3 offspring)) and **g** basal glucose levels in these animals ($n = 20$ (control) and 15 (mT3 offspring)). **h** Glucose i.p. tolerance test (left panel) with area under the curve (AUC, $n = 20$ (control) and 15 (mT3 offspring), right panel). **i** Insulin i.p. tolerance test of control and mT3 offspring ($n = 10$ per group). All values are mean ± SEM with individual values in circles. *$p < 0.05$ and **$p < 0.01$ for maternal T3 treatment. Biological replicates were obtained from 3 to 4 litters per group. Statistical details are provided in Supplementary Data 1. Source data are provided as a Source data file.

addition to the iBAT hyperactivity, we also observed reduced tail temperature even when normalized to body temperature (Fig. 5g), suggesting that the animals have higher peripheral vasoconstriction to minimize heat loss. In contrast to the iBAT phenotype, this effect

was also observed in the 4-week male animals at weaning (Supplementary Fig. 3g).

As this condition of elevated iBAT thermogenesis and enhanced peripheral vasoconstriction resembles the classic

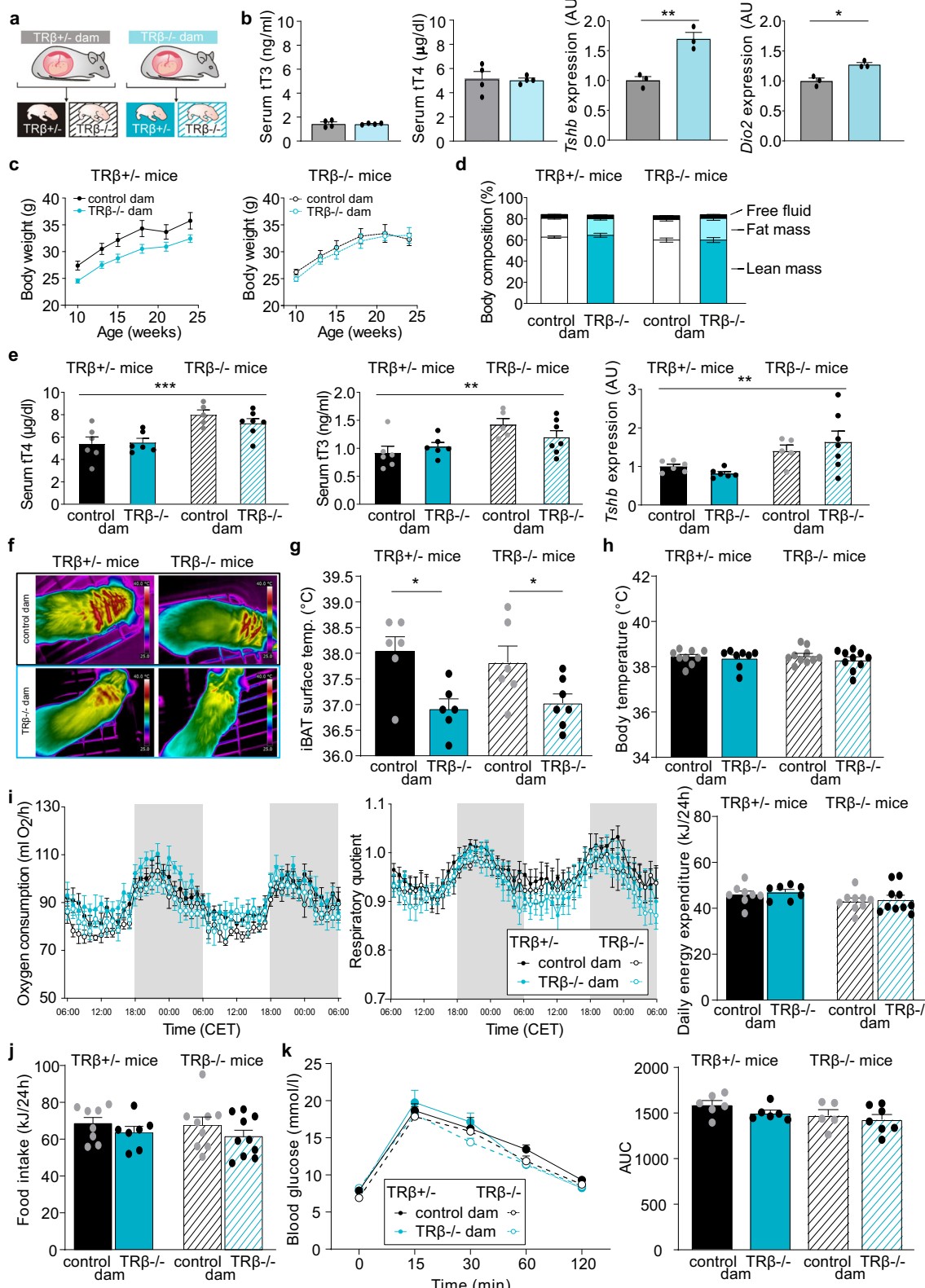

response to cold, we hypothesized that when those animals were exposed to strong cold, the control animals would perform a similar adaptation and the effect would disappear. And indeed, a prolonged cold exposure to 10 °C for 4 weeks, which lead to maximal BAT recruitment (Fig. 5h) and increased serum T3 and T4 with no differences between the groups (Supplementary Fig. 3h), caused strong iBAT activation and tail vasoconstriction in both groups,

thus normalizing the initial difference (Fig. 5i). Interestingly, this massive metabolic activation also reversed the glucose sensitivity phenotype (Fig. 5j); however, a moderately improved insulin sensitivity remained in the mT3 offspring even at this temperature (Fig. 5k). Most remarkably, the better prepared iBAT seemed to reduce the strain on the other thermogenic tissues, as both iWAT and *M. soleus* weight remained higher in the offspring of T3-treated

**Fig. 4 | Role of maternal thyroid hormone receptor β. a** Breeding scheme of the TRβ knockout (TRβ-/-) and control dams. **b** Serum total T3 and total T4 concentrations and pituitary *Tshb* and *Dio2* mRNA expression of control and TRβ-/- dams during pregnancy ($n = 4$ per group for serum, $n = 3$ per group for mRNA). **c** Growth curve of control (left) or TRβ-/- (right) male offspring born by control (black) or TRβ-/- dams (blue). Control offspring: $n = 9$ (from TRβ+/- dam) and $n = 10$ (from TRβ-/- dam), TRβ-/- offspring: $n = 7$ (per group). **d** Body composition with free fluid, fat and lean mass in these animals at the age of 4–5 months (Control offspring: $n = 6$ per group, TRβ-/- offspring: $n = 5$ (from TRβ+/- dam) and $n = 7$ (from TRβ-/- dam)). **e** Serum total T3 and total T4 concentrations and pituitary *Tshb* mRNA expression of control (two left columns in each panel) or TRβ-/- (two right columns) males born by control (black) or TRβ-/- mothers (blue) at the age of 5–6 months (Control offspring: $n = 6$ per group, TRβ-/- offspring: $n = 5$ (from TRβ+/- dam) and $n = 7$ (from TRβ-/- dam)). **f** Representative infrared pictures of the iBAT thermogenesis in these animals with **g** quantification of iBAT surface temperature (Control offspring: $n = 6$ (from TRβ+/- dam) and $n = 6$ (from TRβ-/- dam), TRβ-/- offspring:

$n = 6$ (from TRβ+/- dam) and $n = 7$ (from TRβ-/- dam)) and **h** body temperature at the age of 4–5 months (Control offspring: $n = 9$ (from TRβ+/- dam) and $n = 8$ (from TRβ-/- dam), TRβ-/- offspring: $n = 10$). **i** Oxygen consumption (left) and respiratory quotient (right) over the course of 2 consecutive days. Gray shadows indicate night-time activity period (Control offspring: $n = 8$ (from TRβ+/- dam) and $n = 7$ (from TRβ-/- dam), TRβ-/- offspring: $n = 9$ (from TRβ+/- dam) and $n = 10$ (from TRβ-/- dam)). **j** Food intake in those animals over 24 h (Control offspring: $n = 8$ (from TRβ+/- dam) and $n = 7$ (from TRβ-/- dam), TRβ-/- offspring: $n = 9$ (from TRβ+/- dam) and $n = 10$ (from TRβ-/- dam)). **k** Glucose i.p. tolerance test with area under the curve (AUC, control offspring: $n = 6$ per group, TRβ-/- offspring: $n = 5$ (from TRβ+/- dam) and $n = 7$ (from TRβ-/- dam)). All values are mean ± SEM with individual values in circles. *$p < 0.05$, **$p < 0.01$ and ***$p < 0.001$ for maternal TRβ-/- genotype. Biological replicates were obtained from 3 to 4 litters per group. AU arbitrary units. Statistical details are provided in Supplementary Data 1. Source data are provided as a Source data file.

mothers upon 10 °C (Fig. 5l), indicating that the cold exposure did not consume these tissues as much in the animals as compared to controls.

### T3 induced changes in maternal metabolites

To identify possible candidates mediating the maternal effect, we sacrificed T3-treated pregnant dams and controls at GD17 and performed mass spectrometry-based metabolomics of their serum, resulting in 116 identified metabolites (Supplementary Data 2). The principal component analysis showed distinct clustering of both groups (Fig. 6a). Univariate analysis revealed 5 metabolites with decreased serum levels in the T3-treated dams, and 14 that were increased (Fig. 6b, corrected for multiple testing). Unbiased clustering based on these 19 metabolites resulted in a clear distinction of the two groups (Fig. 6c). Interestingly, choline, a metabolite that participates in the synthesis of *S*-adenosylmethionine (SAM), which is a major methyl donor required for epigenetic DNA methylation[33], was elevated in the serum of T3-treated dams.

Taken together, our data demonstrate that maternal hyperthyroidism reprograms the offspring's handling of body temperature regulation to elevated iBAT thermogenesis and enhanced peripheral vasoconstriction, resembling the adaptations occurring after cold exposure. This effect is observed already early after birth and requires maternal TRβ, as it is reversed in the respective offspring of TRβ-/- mothers.

## Discussion

In this study, we analyzed the effects of maternal hyperthyroidism on fetal programming, revealing alterations in glucose tolerance and iBAT thermogenesis in the next generation. The effects on iBAT activation were specifically mediated by maternal TRβ, as they were opposite in male offspring of dams lacking this receptor. Our results therefore for the first time connect maternal TRβ function to a defined thermoregulatory phenotype in the offspring, demonstrating an important role for this nuclear receptor in fetal programming.

### Effects on thyroid hormone metabolism

It is well established that maternal and offspring TSH or fT4 are positively correlated in humans[12], therefore one would have expected lower TSH levels in the offspring of hyperthyroid mothers, similar to what observed in unaffected infants of mothers with RTHβ[33,34]. In contrast to previous studies, which showed 2-fold elevated serum TSH in the presence of normal T3 and T4[30,35], we did not observe an effect of maternal hyperthyroidism or TRβ knockout on offspring serum T3, T4 or pituitary *Tshb* mRNA expression. Likewise, markers of peripheral thyroid hormone action such as hepatic *Dio1* were unaffected, in line with the normal tissue T3 and T4 content reported in earlier studies[36]. The differences may reflect

an age effect, given that in adulthood, offspring of RTHβ mothers also displayed normal baseline serum TSH, T3 and T4[30,34], but a reduced sensitivity of TSH to thyroid hormone – an effect which seems to be transmitted epigenetically along the male line[34]. This may be mediated by *Dio3* DNA methylation – however, we did not find any effect of maternal thyroid hormone or lack of TRβ on *Dio3* mRNA expression in pituitary or iBAT of our animals.

### Effects of maternal hyperthyroidism on male glucose tolerance

Most remarkably, we observed that maternal T3 treatment caused an improved glucose tolerance selectively in the male offspring. This sexual dimorphism is not surprising, as it was observed for several metabolic parameters previously[37,38]. As the placental differences between male and female fetuses have been implicated in this process[39,40], it is tempting to speculate that the presence of the thyroid hormone inactivating enzyme DIO3, which is regulated in a sexual dimorphic manner[41,42], could cause maternal T3 levels to affect male and female fetuses differently.

Interestingly the glucose tolerance phenotype in the offspring is likely not connected to the iBAT phenotype, as the latter was present in both sexes. Moreover, no change was observed in glucose tolerance of the offspring of TRβ knockout mothers, although their iBAT activity was markedly reduced, and genes involved in iBAT glucose handling were not altered consistently between the animals of the different groups. This is congruent with our observation of normal metabolic rate in all offspring, as altered iBAT thermogenesis is usually compensated by thermoregulatory adaptations in other tissues[21].

### Distinct physiological advantages of the iBAT programming upon cold

The offspring of T3-treated mothers showed increased iBAT temperature in the presence of reduced tail temperature in both sexes, which is reminiscent of the response to cold. However, neither body temperature nor energy expenditure were affected, suggesting that the offspring have a different approach to maintaining their body temperature at room temperature as compared to controls, which shifts the overall thermogenic burden primarily to the iBAT. This altered fetal programming to a prepared iBAT seems to have distinct advantages upon actual cold exposure: (i) early after birth at P5, the mice did not cool out as quickly as the controls, (ii) upon weaning temporarily increased iBAT thermogenesis was required in the controls to counteract the separation from the mother, which was not observed in the offspring of T3-treated mothers, and (iii) in adulthood the prolonged 10 °C cold exposure seems to have put a reduced burden on iWAT and muscle as alternative source for thermogenesis[43–45], as their mass was better maintained. Interestingly, at room temperature, this resetting towards iBAT thermogenesis does not affect overall

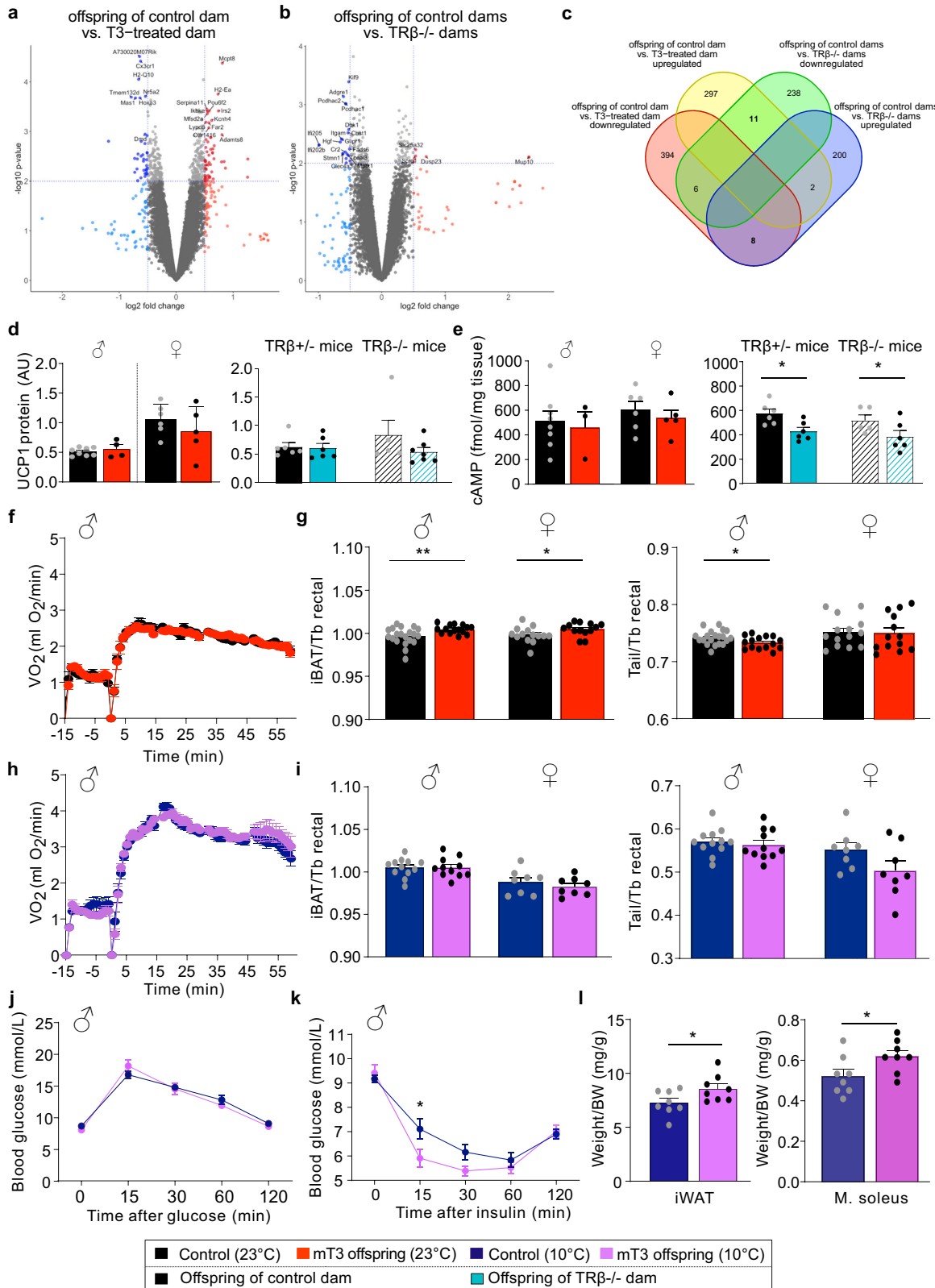

metabolic rate, suggesting that other thermogenic mechanisms e.g. in muscle adapt[46], as expected from previous studies[21].

## Molecular mechanisms of the iBAT programming

The observed shift to iBAT thermogenesis in offspring of T3-treated mothers was interestingly not caused by improved BAT recruitment, as evidenced by a normal NE response. Moreover, iBAT gene expression

was largely unaltered, suggesting that the reprogramming occurs at the central rather than the peripheral level, which is supported by the altered tail vasoconstriction. As we observed lower intracellular iBAT cAMP levels in the offspring of TRβ-/- mothers, indicative of lower sympathetic tone[19], the reduced iBAT thermogenesis in these animals is likely also of central origin. Most importantly, despite relying on different mechanisms, the opposite phenotypes in the two groups

**Fig. 5 | Detailed brown adipose tissue phenotyping in the offspring.**
**a** Microarray study of iBAT gene expression comparing adult male offspring of control and T3-treated dams in a volcano plot ($n = 3$–6). **b** Microarray study of iBAT gene expression comparing adult male offspring of control and TRβ knockout dams in a volcano plot ($n = 5$ per group). **c** Venn diagram depicting the number of significantly changed expressed genes in the different conditions. Only genes with an expression of $\log_2 > 5$ and a $|FC| > 1.25$ were included. **d** UCP1 protein levels in all respective groups (males: $n = 8$ (control) and 4 (mT3 offspring), females: $n = 6$ (control) and 5 (mT3 offspring); control offspring: $n = 6$ (from TRβ+/- dam) and $n = 6$ (from TRβ-/- dam), TRβ-/- offspring: $n = 5$ (from TRβ+/- dam) and $n = 7$ (from TRβ-/- dam). **e** cAMP levels in iBAT of all respective groups (males: $n = 8$ (control) and 3 (mT3 offspring), females: $n = 6$ (control) and 5 (mT3 offspring); control offspring: $n = 6$ (from TRβ+/- dam) and $n = 6$ (from TRβ-/- dam), TRβ-/- offspring: $n = 5$ (from TRβ+/- dam) and $n = 6$ (from TRβ-/- dam). **f** Oxygen consumption of adult male offspring of control (black) and T3-treated dams (red) in response to a single injection of norepinephrine at time point 0 ($n = 10$ (control) and 8 (mT3 offspring)). **g** iBAT and tail surface temperature in the animals normalized to the respective body temperature (males: $n = 20$ (control) and 15 (mT3 offspring), females: $n = 14$ (control) and 13 (mT3 offspring)). **h** Oxygen consumption of adult male offspring of control (dark blue) and T3-treated dams (purple) kept for 2–3 weeks at 10 °C in response to a single injection of norepinephrine at time point 0 ($n = 6$ per group). **i** iBAT and tail surface temperature in these animals normalized to the respective body temperature after 3 weeks of cold exposure (males: $n = 12$ (control) and 11 (mT3 offspring), females: $n = 8$ (control) and 8 (mT3 offspring)). **j** Glucose and **k** insulin tolerance test in these animals (control: $n = 11$ (GTT)/12 (ITT) and mT3 offspring: $n = 10$ (GTT)/10 (ITT). **l** Weight of inguinal white adipose tissue (iWAT) and soleus muscle (*M. soleus*) in the animals after 4 weeks of 10 °C exposure ($n = 8$ per group). All values are mean ± SEM with individual values in circles. *$p < 0.05$ and **$p < 0.01$ for maternal T3 treatment. Biological replicates were obtained from 3 to 4 litter per group (mT3 offspring (23 °C) and TRβ offspring) or 4 to 6 litters per group (mT3 offspring (10 °C)). AU arbitrary units. Statistical details are provided in Supplementary Data 1 and Supplementary Table 1. Source data are provided as a Source data file.

demonstrate that (i) maternal TRβ is required for the fetal programming effect of T3 and (ii) that the T3 action occurs largely in the mother rather than the embryo. The effect therefore differs from a recent observation in mice with a BAT specific inactivation of DIO3, where premature exposition of the fetal BAT to elevated T3 levels also lead to an increased preparedness for cold, but those mice were lighter and had substantial changes in iBAT gene expression albeit normal UCP1 in adulthood[29,47].

The observation of an altered central programming by elevated maternal thyroid hormone is not unusual and has been demonstrated previously in mice with a mutation in *Dio3*: Here the overexposure of the offspring to thyroid hormone caused different DNA methylation of several genes involved in early development of the brain, linking maternal thyroid hormone signaling to epigenetic alterations in the offspring central nervous system[48]. Our findings of elevated serum choline in the T3-treated pregnant dams provide a possible molecular mechanisms, given that maternal choline is required for neuronal development[49] and has been implicated previously in the fetal programming of the brain, including the central control of energy homeostasis[50,51].

## Clinical relevance

Given the relatively high incidence of thyroid hormone disorders in pregnant women[52], the consequences of altered maternal thyroid hormone signaling for fetal programming cannot be underestimated in the context of public health. This is especially relevant for metabolic disorders, which are rising on a global scale, and the reasons for their development are still far from being understood, particularly as genetic contributions are relatively minor[53]. Therefore, the data from our study, connecting maternal endocrinology via a specific nuclear receptor to fetal programming of glucose tolerance and BAT activity may contribute to a better understanding of the developmental origins of health and disease. However, additional studies in female offspring will be required to characterize further the sexual dimorphism of the effect. Moreover, they further advocate for the implementation of a routine thyroid screening in pregnant women, which is still controversially discussed to date[13,54,55].

## Methods
### Animal procedures

Hyperthyroidism during pregnancy was induced in wildtype females (Charles River, Germany) at the age of three to four month with 0.5 mg/L T3 (3,3′,5-Triiodo-L-thyronine (T6397, Sigma Aldrich, Germany) in drinking water with 0.01% BSA. Treatment started at the day of a positive plug check and was continued until GD17. Offspring of mothers that received T3 during pregnancy are referred to as m(maternal)T3 offspring. Litters were reduced to a maximum of 8 pups per group at postnatal day 1 or 2. Mothers with fewer than 6 pups per litter were excluded from further analysis to avoid differences in maternal care.

To decipher effects related to maternal thyroid hormone receptor β signaling, offspring obtained from TRβ knockout (TRβ-/-) and control females (TRβ+/-) were compared. Female mice from our colony at the Gemeinsame Tierhaltung (GTH) Lübeck were mated with heterozygous males (TRβ+/-) to generate TRβ knockout (TRβ-/-) and control (TRβ+/-) offspring from both groups. Noteworthy, animals heterozygous for the TRβ knockout allele display similar phenotypically characteristics as wildtype animals[30,32].

Unless otherwise stated mice were kept on a constant 12-hour light/12-hour dark cycle at $23 \pm 1$ °C and 40–60% humidity with free access to food and water (breeding diet #1314 from Altromin, Germany, 14% fat, 27% protein, 59% carbohydrates, metabolized energy: ~3.339 kcal/g or 13.97 kJ/g). All animals were on a C57BL/6NCrl background strain, which was controlled by genetic typing regularly. The 10 °C control and mT3 offspring were transferred to 10 °C at the age of 10 weeks and were kept for 2 weeks before analyses. The respective ages of the animals are provided in the results or figure legends. Animals were monitored daily, euthanized using carbondioxide or isoflurane in combination with cervical dislocation, and procedures were approved by the Ministerium für Energiewende, Klimaschutz, Umwelt und Natur MEKUN Schleswig-Holstein, Germany.

At different time points during development, infrared pictures (T335, FLIR, Sweden) from the interscapular brown adipose tissue (iBAT) area and tail surface were taken, while animals were freely moving on a cage grid[56]. Pictures at postnatal day 5 were taken 2 min after removing the newborns from the mother's nest. Infrared thermography was performed at the respective housing temperatures (23 or 10 °C) and analyzed with FLIR Tools/Tools+ software (FLIR). Immediately after the infrared pictures, the rectal body temperature was measured (BAT-12, Physitemp, USA). Body composition was analyzed using Minispec LF110 and Minispec Plus Software 6.0 (Bruker, USA). At the day of sacrifice, organs were collected, weighed and snap-frozen at −80 °C, whereas serum was stored at −20 °C.

### Glucose (ipGTT) and insulin (ipITT) tolerance test

Glucose tolerance test (ipGTT) in pregnant mice was performed at gestational days 14–15, whereas the adult offspring were tested at more than 2 months of age or after 3 weeks of cold exposure. Animals were fasted for 6 h before intraperitoneal injection of glucose (2.0 g/kg body weight). Blood glucose concentrations were measured before and 15, 30, 60 and 120 min post injection in blood drawn from the tail vein using a commercially available glucometer (AccuCheck, Aviva, Germany). The insulin tolerance test was performed after 2 days of

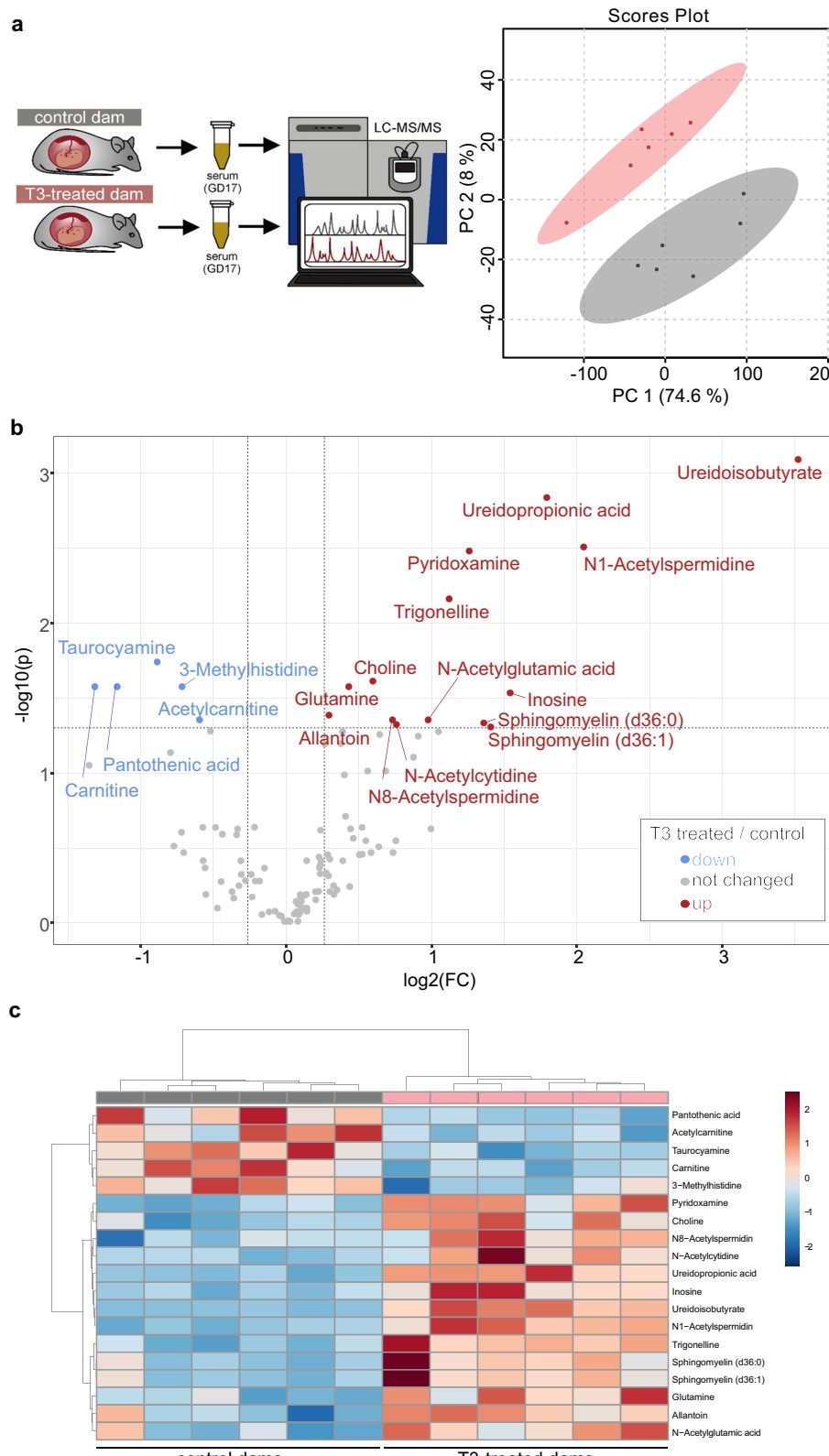

**Fig. 6 | Metabolomics in T3-treated mothers. a** Methodological overview and principal component analysis of metabolomics dataset, including 116 metabolites in serum samples from mothers treated with T3 during pregnancy and controls as analyzed by mass spectrometry ($n = 6$ per group). **b** Volcano plot depicting fold-change and significance level corrected for multiple testing in those samples. **c** Heatmap generated by unbiased clustering of samples based on the significantly changed metabolites in these samples. GD17 gestational day 17, LC-MS/MS liquid mass spectrometry, PC principal component. Statistical details are provided in Supplementary Data 2.

recovery. After an intraperitoneal injection of insulin (0.5 U/kg body weight, NovoRapid Penfill 100E/ml, Novo Nordisk) blood glucose concentrations were determined as described above.

## Indirect calorimetry

Daily energy expenditure was measured via indirect calorimetry using an open respirometry system and a temperature-controlled (23 ± 0.5 °C) chamber (TSE PhenoMaster, TSE systems, Germany). Oxygen consumption, carbon dioxide production, food and water intake, respiratory quotient (RQ = carbon dioxide produced/oxygen consumed) and activity were measured simultaneously in 20 minute intervals. DEE was calculated using the RQ and the caloric equivalents given by[57]. Resting metabolic rate (RMR) at 30 °C was determined in the inactive phase (lights on), over a 1-h interval of consistently low oxygen consumption rates and without physical activity in animals that were fasted for six hours. Maximum non-shivering thermogenesis was determined by injecting fasted animals with norepinephrine (NE, 1 mg/kg arterenol, Sanofi, Germany) at 23 °C. Oxygen consumption and carbon dioxide production were measured in 20 s intervals with an open flow-through respirometry set up with one reference and one measurement channel (CaloBox, PhenoSys GmbH). Data were analyzed with TSE PhenoMaster software (V5.8.1, V6.2.0 and V6.5.3, TSE Systems, Germany) or CaloBox software (PhenoSys GmbH, Germany).

## Pup retrieval test

The mother was removed from the cage and the litter was placed on the opposite side of the nest to measure the latency to retrieve the pups back to the nest. The mother was then placed back in the cage and the time to first sniff the litter and retrieve the pups was measured. Mothers were either observed until all pups were carried to the nest or for a maximum of 3 min.

## Serum T3 and T4

Commercially available ELISAs were used to determine serum levels of total T4 (EIA-1781, DRG Diagnostics, Germany) and total T3 (DNOV053, NovaTec Immundiagnostica GmbH, Germany). Briefly, 25 μl (tT4) or 50 μl (tT3) of serum or standard were added to a well, followed by 100 μl of horseradish peroxidase (HRP)-linked conjugate solution. The plate was then incubated at room temperature for 1 h and washed 5× with 100 μl of washing buffer before adding the HRP substrate 3,3′,5,5′-Tetramethylbenzidine (TMB). The reaction was stopped with 100 μl of 0.5 M sulfuric acid and extinction was measured with a spectrophotometer SPECTROstar Nano Microplate Reader at a wavelength of 450 nm.

## cAMP assay

The intracellular cAMP levels of iBAT tissue samples were measured using a commercially available ELISA kit (RPN225, GE Healthcare, UK). For this purpose, dry-ice frozen iBAT samples were homogenized in 250 μl trichloroacetic acid at 4 °C. After centrifugation (2000 × $g$, 15 min, 4 °C), the supernatant was washed 5x with water-saturated diethyl ether before drying it under a stream of nitrogen. The remaining pellet was dissolved in 250 μl assay buffer before the ELISA was conducted according to the manufactures instructions. Finally, absorbance was measured using a SPECTROstar Nano Microplate Reader at a wavelength of 450 nm and the results obtained were adjusted to the tissue weight used (10–20 mg).

## Gene expression analysis

For gene expression analysis, RNA was isolated using QIAGEN RNeasy Kits (QIAGEN, Germany). Subsequent cDNA synthesis was carried out using the Molecular Biology RevertAid strand cDNA Kit (Thermo Fisher Scientific, Germany) and anchored oligo (dT)18 primers. qPCR analysis was performed using SYBR Green PCR

Master Mix (Roche, Germany) and QuantStudio Applied Biosystems (Thermo Fisher Scientific, Germany). The most stable reference genes were determined with geNorm version 3.4. (http://medgen.ugent.be/genorm/) and gene expression levels were normalized to the mean of those reference genes (*peptidylprolyl isomerase D (Cyclophilin D)*: frw TCA CAA CAG TTC CGA CTC CTC, rev ACC TCT ACA TTT TCA AGC GTC C and *hypoxanthine-guanine phosphoribosyltransferase (Hprt)*: frw GCA GTA CAG CCC CAA AAT GG and rev AAC AAA GTC TGG CCT GTA TCC AA). PCR efficiency was corrected by the calculation of standard curves for each primer pair. Primer sequences *Dio2*: frw ATG GGA CTC CTC AGC GTA GAC, rev ACT CTC CGC GAG TGG ACT T and *Tshb*: frw GGG CAA GCA GCA TCC TTT TG, rev GTG TCA TAC AAT ACC CAG CAC AG. Microarrays were conducted by Atlas Biolabs (Berlin, Germany) on GeneChip Clariom S arrays (Affymetrix/Thermofisher, Germany). Raw microarray data (CEL files) were preprocessed using the oligo R package (v1.50.0) to correct the background and perform quantiles normalization using the Robust Multichip Average (RMA) algorithm[58,59]. Additional quality control was conducted using the Biobase (v2.46) and oligo R packages[60]. An outlier was detected for the T3 samples using the outlier detection test of the R package arrayQualityMetrics (v3.42)[61] and removed. Genes expressed below a threshold of 4 were filtered out from the normalized data based on their median transcript intensities. Differentially expressed genes (DEGs) were screened using the Linear Models for Microarray data (LIMMA) method implemented in the R package limma (v3.42)[62]. Array weights were considered in the analysis to increase the signal-to-noise ratio. The threshold for identification of DEGs was set as $p$ value < 0.01 and |$\log_2$ fold change (FC)| ≥ 0.5. For the Venn-Diagram only genes with an expression of $\log_2$ > 5 and a |FC| > 1.25 were considered.

## Western blot UCP1/OxPhos

For protein isolation snap-frozen tissue of iBAT and muscle were homogenized in RIPA buffer (150 mM NaCl, 50 mM Tris-HCl pH 7.5, 0.1% w/vol SDS, 0.5% w/vol sodium deoxycholate, 1% vol/vol Nonidet P40, 1 mM EDTA, 1 mM EGTA, 2.5 mM sodium pyrophosphate, 1 mM $NaVO_4$ and 10 mM NaF) and freshly added protease inhibitors (Roche Diagnostics GmbH, Germany). Protein concentrations were determined using a bicinchoninic acid assay (BCA, Sigma, Germany). For immunological detection, 20 μg of iBAT protein were separated on a 12% SDS polyacrylamide gel (BioRad Laboratories, Germany) and transferred onto a polyvinylidene fluoride membrane (Merck Millipore, Germany). The membranes were probed with a rabbit anti-UCP1 antibody (customized rabbit antibody raised against UCP1[63], or a mouse anti-total OxPhos antibody cocktail (#45-8099, Invitrogen Germany), followed by a peroxidase-conjugated secondary antibody (goat anti-rabbit-IgG, #P0448, DAKO, Denmark or goat anti-mouse-IgG, #P0447, DAKO, Denmark). The antigens were visualized using an ECL Plus system (Chemi Doc Touch, BioRad). Band intensity was quantified using Image Lab™ software (BioRad) and data were normalized to total protein load.

## Metabolomics

A Dionex Ultimate 3000 RS LC-system coupled to an Orbitrap mass spectrometer (QExactive, ThermoFisher Scientific, Germany) equipped with a heated-electrospray ionization (HESI-II) probe was used. All solvents were of LC-MS grade quality (Merck Millipore, Germany).

For metabolic profiling, 400 μl acetone/acetonitrile/methanol (1:1:1, v/v/v), containing 2.5 μM Metabolomics Amino Acid Mix Standard (Cambridge Isotope Laboratories, United States), were added to 100 μl serum[64]. After incubation and centrifugation, supernatants were dried under vacuum and reconstituted in 50 μl methanol/acetonitrile (1:1, v/v). Metabolites were separated on a

SeQuant ZIC-HILIC column (150 × 2.1 mm, 5 μm; Merck Millipore, Germany) using 5 mM ammonium acetate (Sigma Aldrich, Germany) as eluent A and acetonitrile/eluent A (95:5, v/v) as eluent B. The gradient elution was set as follows: isocratic step of 100% B for 3 min, 100% B to 60% B in 15 min, held for 5 min, returned to initial conditions in 5 min and held for 5 min. Flow rate was 0.5 ml/min. Data acquisition with data dependent $MS^2$ scans (top 10) was performed. Metabolites were identified based on exact mass, retention time, fragmentation and isotopic pattern using Compound Discoverer 3.1 (ThermoFisher Scientific, Germany), including an in-house $MS^2$ library[64] as well as the online library mzCloud. Pooled samples at 5 concentrations were used as quality controls and for normalization. Normalized areas under the curve (AUC) were analyzed using MetaboAnalyst 5.0[65]. Univariate analysis was performed based on an unpaired Student's t-test and p-values were adjusted for false discovery rate (FDR) by the Benjamini-Hochberg procedure. Fold-change threshold was set to 1.2 and p value (adjusted) to ≤0.05. Cluster analysis was based on Euclidean distance measure and Ward clustering algorithm.

### Statistical analysis
Results of the T3-treated dams and mT3 offspring were analyzed with an unpaired Student's t test with Welch's correction, whereas 2-WAY ANOVA with sex and maternal treatment as factors was used to analyze the results of offspring from TRβ deficient mothers (Graphpad Prism Version 7-9). GraphPad 7.0 was used to identify and exclude significant outliers (ROUT method, $Q = 1\%$). Significant results were subsequently analyzed with a post hoc test, correcting for multiple comparisons. All statistical results are provided in Supplementary Data 1. The data are presented as mean ± SEM with individual values in circles.

### Reporting summary
Further information on research design is available in the Nature Portfolio Reporting Summary linked to this article.

## Data availability
Source data are provided with this paper or have been deposited in the ArrayExpress database at EMBL-EBI under accession number E-MTAB-12587 for the microarray study, and at MassIVE under accession number MSV000091042 (https://massive.ucsd.edu/ProteoSAFe/dataset.jsp?task=53dcd590d2b94f2aa318f5f9b2ae64c5) for the metabolomics data respectively. Source data are provided with this paper.

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

## Acknowledgements

We thank the staff of the GTH of the University of Lübeck for technical assistance. This work was supported by grants of the Deutsche Forschungsgemeinschaft (MI1242/2-2 and 3-2 Heisenberg Program, Mi1242/4-1 within SPP1629 "Thyroid TransAct" and Project-ID 424957847 - TRR296 "LocoTact" to J.M.; Project-ID 434396546 to R.O.), and the Medical Faculty of the University of Lübeck (J14-2018) to R.O. L.H., M.P., S.G., and K.J. were students of the GRK1957 "Adipocyte-Brain-Crosstalk", B.K. is a student of the TRR296 "LocoTact" integrated research training group.

## Author contributions

R.O., L.H., M.P., B.K., J.I., S.G., J.R., and K.J. performed experiments; R.O., A.H., O.J., K.K., and J.M. analyzed and interpreted the data; R.O. and J.M. conceptualized and designed the study; J.M. drafted the manuscript, which was substantially edited and approved by all authors.

## Funding

## Competing interests

The authors declare no competing interests.
