## [Peer Review File · Nature Communications]

Maternal thyroid hormone receptor β activation in mice sparks brown fat thermogenesis in the offspringReviewers' comments:

Reviewer #1 (Remarks to the Author):

This is a very interesting article written by Oelkrug et al. which links maternal hyperthyroidism, the thyroid hormone nuclear receptor beta and thermoregulation in the offspring. This is a novel finding – the first one linking the TR nuclear with offspring thermogenesis and sex-specific improvement in offspring glucose tolerance.

However, the data in the manuscript is complicated by the fact that the offspring appear to have been studied only once at 25 weeks of age a rather old age (25 weeks is mentioned in the text in the discussion but 15 weeks is shown in Figure 1). It is not explained why the mice were studied at this age. Previous studies, especially in rats but also in human demonstrate long-term suppression of the hypothalamic-pituitary axis which is not seen here, presumably due to when the mice were studied. The fact that the mice were studied at only 1 age leaves open that the phenotype is transient.

Specific Comments:

1. It would likely be important to show this phenotype at one other time point especially earlier in development. For example this may be a transient phenotype not a persistent phenotype.
2. It also remains unclear what the mechanism may be ie why does the iBAT remain more active or why is it reprogrammed. For example are epigenetic marks different in treated versus untreated offspring. No attempt at determining a mechanism is taken forward from this point of view.
3. The opposite effect is seen in the RTH offspring suggest TRbeta specific mechanism. However, it is not clear if this is mediated centrally or in brown fat directly. This probably should be tested.
4. A small point: it is not clear at what age were the RTH offspring studied at?
5. The discussion is very difficult to understand. The clinical extrapolations to humans are overstated.

Reviewer #2 (Remarks to the Author):

This study was designed to determine the programming effects of maternal thyroid hormone on energy metabolism in offspring later life. However, the research was poorly designed, and the manuscript is tough to read. The following are my main concerns.

1. Lacking data for most conclusions. For example, what causes the increase in temperature of iBAT without a change in UCP1 protein? How is insulin sensitivity programmed?
2. No detailed information was provided for some studies. The term "endogenous hyperthyroidism" was mentioned several times in this ms. What is it? Was T3 treatment applied to the dams of Figure 4? It is not clear why used THb knockout mice. Studying the direct effect of T3 in the fetal compartment?
3. It sounds that thyroid hormones can pass through the placenta in the introduction section. It is logical to study the direct effects of increased T3 in fetal development. It is odd to know that there was no change in fetal blood T3 and TSH.
4. What are the effect of T3 treatment and TRb deletion on maternal metabolism? This should be the key information of this ms.
5. Some of the discussion is confusing. For example, RTHb was discussed, but there is no RTHb data in the result section. Most of the conclusions in this section are speculative, lacking data support.

Reviewer #3 (Remarks to the Author):

This manuscript reports on the maternal effects of pharmacological and genetic manipulations in the endocrine thyroid hormone system on the adult offspring. The main results are that T3 treatment of pregnant dams leads to hyperactivity of BAT in the adult offspring, and genetic ablation of TRbeta in dams leads to hypoactivity of BAT in adult offspring. These results are potentially of interest to better understand how maternal thyroid status programs offspring metabolism and energy homeostasis. As highlighted in the introduction, in humans maternal effects of hypo- and hyperthyroidism on offspring phenotypes are well known, but the underlying mechanism remain unclear. The experimental design chosen in the present study enables to dissect whether maternal hyperthyroidism depends on the presence of TRbeta in the dam or in the pups.

Although this work is of interest, however, the experimental evidence from mouse models does not go beyond describing an interesting phenomenon, but falls short in delivering mechanisms.

Major points of criticism

The main conclusion is that BAT hyperactivity in offspring is mediated by T3, dependent on TRbeta expression in the dams. The authors therefore postulate an indirect mechanism, involving an unknown circulating factor derived from a TRbeta dependent tissue, like liver, that crosses the placental barrier and conveys programming of BAT. No attempts to identify the unknown factor were undertaken.

The present report consolidates the already known involvement of maternal TRbeta in programming of offspring. Maternal programming effects have been reported for humans lacking functional TRbeta (RTHbeta), as referenced by the authors in the introduction (17,18).

The introduction is unbalanced as it almost exclusively addresses the negative effects of maternal hypo- and hyperthyroidism on fetal development and the health status of the offspring in humans. Regarding the sole focus of this paper on experimental work in mice and BAT, the introduction fails to introduce important facts on the role of thyroid hormones for BAT physiology, and what we know about hypo- and hyperthyroidism in rodents in respect to maternal effects. Most of the human relevance part should move to the discussion.

In addition, there is a major discrepancy between the observed programming effect on BAT in mice and the maternal effects of altered thyroid hormone activity in humans on neuronal / psychiatric phenotypes which is not sufficiently addressed by the authors.

Negative data certainly are of indispensable value. However, in this manuscript the reader is exposed to a large amount of mostly negative data. It remains unclear why it is important to include all these negative findings in the main body of the manuscript, and not move them to the supplement.

How was the measurement of BAT temperature validated? Was the interscapular area shaved prior to the measurement? Was a reference temperature defined at a more posterior position of the mouse?

On page 6 of results, after summarizing the T3 treatment findings, they do not explain properly why they chose to study a possible contribution of maternal TRbeta. The potential mechanistic link between the opposing actions of T3 treatment and genetic ablation of TRbeta in dams on BAT activity in adult offspring is missing. T3 treatment of dams increased BAT temperature (figure 3C) whereas offspring born from TRbeta KO dams showed decreased BAT temperature (figure 6C). However, during fetal development the latter were exposed to elevated maternal T3 levels, alike

T3 treatment. On page 6 we read that endogenously hyperthyroid TR-beta KO females were mated, but thyroid hormones were not measured in TR-beta KO dams. Is the T3 elevation comparable to the T3 treatment? Why was there differential response to a similar endocrine stimulus? This discrepancy should be subject to the discussion.

Alike BAT temperature, BAT cAMP levels were reduced in offspring from TRbeta KO dams. This may be due to reduced sympathetic tone as concluded by the authors, but could also be due to altered expression of adrenergic receptors, G-proteins or adenylylcyclase upstream of cAMP. Notably, cAMP in BAT was not affected by T3-treatment.

On page 9, we read that RMR at thermoneutrality was higher, but this is not supported by the stats (Figure 5E, Supplemental Table), and is in contrast to the description of no difference in RMR on page .

Regarding the energy balance data, duration of measurements for food intake are too short to detect differences between treatment groups, as food intake data normally show more day-by-day variation than energy expenditure recordings.

The presentation of metabolic data is not well structured. It remains unclear whether there are significant maternal effects on metabolic rates of adult offspring, as the applied analyses of metabolic data are not state of the art. First, oxygen consumption and DEE data are presented per mouse (figures 2C-F and 5C,D). These should be directly compared with the respective food & water intake data (figure 2I, 5F), and with the locomotor activity data. Energy in and energy out should be expressed as energy units (kcal or kJ). While DEE, food & water consumption are expressed per mouse, why did the authors choose to show mass-specific ratios for RMR (figures 2G,H, and 5E). Several reviews address the problems of data analyses, and the use of mass-specific ratios has been heavily criticized in the past. One solution could be to calculate ratios per gram of lean mass. However, ANCOVA or GLM based stats are recommended. One solution would be to use the CalR platform for state of the art data analyses (<https://calrapp.org/>).

Why do body composition percentages not add up to 100%? Small differences in body composition may be detectable / verifiable by using proper statistical methods (ANCOVA) instead of mass-specific ratios (percentages in figures 1E and 4D,E).

Did T3 treatment or the genotype of dams have an effect on duration of gestation. Did these interventions delay or accelerate the timing of parturition?

We thank the reviewer and the Editor for the constructive criticism. As you can see, we have substantially revised the manuscript and added several new data, e.g. on the pregnant females, microarray data on the offspring BAT of T3 treated and TRb knockout mothers, a cold exposure study and metabolomics in the pregnant females. We apologize for the delay in the resubmission, which was partially caused by a COVID19 lockdown of our animal facility. We hope that the reviewers will appreciate the additional work and that all concerns have been satisfyingly addressed.

Reviewer #1 (Remarks to the Author):

This is a very interesting article written by Oelkrug et al. which links maternal hyperthyroidism, the thyroid hormone nuclear receptor beta and thermoregulation in the offspring. This is a novel finding – the first one linking the TR nuclear with offspring thermogenesis and sex-specific improvement in offspring glucose tolerance.

However, the data in the manuscript is complicated by the fact that the offspring appear to have been studied only once at 25 weeks of age a rather old age (25 weeks is mentioned in the text in the discussion but 15 weeks is shown in Figure 1). It is not explained why the mice were studied at this age. Previous studies, especially in rats but also in human demonstrate long-term suppression of the hypothalamic-pituitary axis which is not seen here, presumably due to when the mice were studied. The fact that the mice were studied at only 1 age leaves open that the phenotype is transient.

Answer: We thank the reviewer for the positive evaluation of our manuscript and the kind and constructive criticism, which we have addressed as outlined below. In short, we have included infrared images of iBAT from offspring of T3-treated and control mothers at two other time points, which clearly show an increase in brown fat thermogenesis already immediately after birth, demonstrating that the phenotype presented is a persistent phenotype.

Specific Comments:

1. It would likely be important to show this phenotype at one other time point especially earlier in development. For example this may be a transient phenotype not a persistent phenotype.

Answer: The reviewer is absolutely correct that a central issue in our study is whether the increased BAT thermogenesis of the offspring exists immediately at birth or develops only later in life. We therefore repeated the study with a new cohort and added IR pictures of the iBAT of male and female mT3 offspring (offspring from mothers treated with T3) at two earlier time points to our manuscript.

The first time point at postnatal day 5 shows an elevation in iBAT temperature already right after birth (Fig 1I, J, see left side). Interestingly, at the second time point around weaning (4 weeks of age), where the mice are separated from the mothers, the BAT temperature is not different from controls

(Suppl. Figure 1E); however, this is caused by temporarily increased BAT thermogenesis in the control animals (from around 38.2°C to 38.6°C), presumably due to the mild cold stress that weaning entails. At the age of 9 weeks, BAT temperature of controls returned to normal and mT3 offspring was again significantly higher (Figure 2E, F). These data have now been included in the manuscript, demonstrating that the effect occurs very early in life and persists.

2. It also remains unclear what the mechanism may be why does the iBAT remain more active or why is it reprogrammed. For example are epigenetic marks different in treated versus untreated offspring. No attempt at determining a mechanism is taken forward from this point of view.

Answer: We agree with the reviewer and have therefore performed an in-depth analysis of the iBAT phenotype (Fig 5). This included a transcriptomic analysis of iBAT samples of mT3 and TRb offspring, which showed minor changes in the iBAT transcriptome as a consequence of the maternal treatment in both groups. However, when the analyses were combined (Fig 5C) to identify genes that could explain elevated BAT thermogenesis in mT3 and reduced iBAT thermogenesis in TRb offspring (i.e. opposite regulations in both groups), only 19 genes were found, and when applying stricter criteria for multiple testing, only 1 gene (a pseudogene of an olfactory receptor) remained, suggesting that the phenotype is likely not caused by epigenetic reprogramming of the iBAT itself.

This was subsequently confirmed by further physiological data, showing lower tail temperature of mT3 offspring at 23°C (Figure 5G right panel, see below), which in the presence of elevated iBAT thermogenesis (Fig 5G left panel) suggests a central cold-like adaptation. To test this in greater detail, we have now included a new cohort of mice that were exposed to 10°C for 4 weeks as adult animals. As expected, this led to a maximal induction of iBAT thermogenesis and a maximal vasoconstriction in the tail of the control mice, causing the initial difference between the groups to disappear (Fig 5H-L).

Most remarkably, however, the preactivated iBAT thermogenesis allowed the mT3 offspring to reduce the strain on the other thermogenic tissues muscle and inguinal white fat, thus better maintaining their tissue mass as compared to the controls (Fig 5L). These new data have been included in the manuscript and suggest that the maternal effect is a general and central alteration of how thermoregulation is regulated to maintain body temperature (more iBAT, more vasoconstriction, thus conserving iWAT and muscle) and likely not a direct reprogramming of the iBAT gene expression itself.

3. The opposite effect is seen in the RTH offspring suggesting TRbeta specific mechanism. However, it is not clear if this is mediated centrally or in brown fat directly. This probably should be tested.

Answer: We agree with the reviewer and have also studied the TRb offspring in greater detail. Here, we observed that the iBAT cAMP levels in those offspring were lowered (Fig 5E), indicative of reduced sympathetic stimulation. Together with the lack of major changes in gene expression (Fig 5B and answer to question 2 above), we conclude that similar to mT3 offspring, the observed phenotype is a central reorganization of thermoregulation rather than a direct BAT effect.

4. A small point: it is not clear at what age were the RTH offspring studied at?

Answer: We apologize for omitting this information. The physiological analyses like IR and TSE were obviously conducted earlier (4-5 months of age) than the tissue analyses (when the animals had been killed, 5-6 months of age). We have now therefore added the specific age of the animals at the time of each experiment to the figure legends.

5. The discussion is very difficult to understand. The clinical extrapolations to humans are overstated.

Answer: We appreciate this comment and have therefore entirely restructured and shortened the discussion. The clinical relevance is now only mentioned in a small paragraph at the end and a rather general recommendation to monitor thyroid status during pregnancy.

Reviewer #2 (Remarks to the Author):

This study was designed to determine the programming effects of maternal thyroid hormone on energy metabolism in offspring later life. However, the research was poorly designed, and the manuscript is tough to read. The following are my main concerns.

Answer: We thank the reviewer for the constructive criticism, giving us the opportunity to improve and clarify our study in a revision. We agree that the way we presented our findings was somewhat confusing, making it difficult to extract the research design and solid conclusions. We have therefore entirely restructured the manuscript, removing many of the non-significant observations and moving other to the supplementary information. We hope that the new layout of the manuscript will make it easier to understand the logic of the study design and allow a better flow of information.

1. Lacking data for most conclusions. For example, what causes the increase in temperature of iBAT without a change in UCP1 protein? How is insulin sensitivity programmed?

Answer: In response to this concern and also the concerns of the other reviewers, we have added an entirely new figure with transcriptomics, norepinephrine stimulation tests and cold exposure (Fig 5) to provide further insight into the change in iBAT thermogenesis.

With regard to the UCP1 levels, we would like to point out that iBAT thermogenesis can in fact be elevated without elevating UCP1 protein levels first and vice versa, as high levels of UCP1 do not automatically translate to higher thermogenesis (see excellent review by Li & Fromme, *IJMS* 2022, <http://dx.doi.org/10.3390/ijms23052406>). Therefore we routinely use infrared thermography to assess iBAT function rather than relying only on molecular

data, as UCP1 expression cannot be directly linked to actual thermogenesis. To test whether UCP1 activation is indeed similar in both groups, we have now performed norepinephrine stimulation tests

in the mT3 offspring, both at room temperature as well as after 10°C cold exposure, a condition with maximal UCP1 induction (Fig 5F and H, see figure above). As expected, the response was almost double at 10°C, but still similar between mT3 and control offspring, thus confirming that the amount of activatable UCP1 in the animals, i.e. the iBAT recruitment, is indeed not different. Together with the finding of low cAMP in the TRb offspring (Fig 5E) and the reduced tail temperature in the mT3 offspring, indicative of accompanying vasoconstriction (Fig 5G) as well the only mildly affected gene expression profile in the iBAT itself (Fig 5A-C), we conclude that the altered thermogenesis is not a tissue specific reprogramming but a central alteration on how body temperature maintenance is governed. This was finally confirmed by the findings after cold exposure, showing a reduced strain on the other thermogenic tissues (muscle and inguinal white adipose tissue), which maintained more mass in the mT3 offspring as compared to controls.

With regard to the insulin sensitivity, we performed an insulin tolerance test in the mT3 offspring (Fig 3I), showing that the response to insulin is entirely comparable, albeit on a lower glucose level background (Fig 3G). The lower glucose level is likely not caused by the iBAT, as there was no change in genes involved in glucose handling (Suppl Fig 3A), and the iBAT and glucose phenotype segregate in female mT3 offspring, which display the elevated thermogenesis but no change in glucose tolerance. Likewise the TRb offspring show the thermogenesis phenotype with no change in glucose tolerance. Interestingly, the glucose tolerance was normalized upon cold exposure, presumably due to the maximal glucose demand in the controls (Fig 5J); however, the insulin sensitivity at 10°C was higher in the mT3 offspring, presumably due to their increased muscle mass (Fig 5L).

2. No detailed information was provided for some studies. The term “endogenous hyperthyroidism” was mentioned several times in this ms. What is it? Was T3 treatment applied to the dams of Figure 4? It is not clear why used THb knockout mice. Studying the direct effect of T3 in the fetal compartment?

Answer: We apologize for not providing a more detailed information on this animal model. The term “endogenous hyperthyroidism” was obviously confusing and has thus been eliminated from the manuscript. Instead, we have measured the activity of the HPT axis in the pregnant dams (Fig 1B for T3 treated dams and Fig 4B for TRb knockout mothers), which now allow a direct comparison of the two models.

The TRb knockout mothers have indeed been used to test whether maternal TRb would be required for the observed phenotype: if removal of this receptor in the mother would reverse the phenotype (as it did in terms of BAT thermogenesis), we could conclude that it is necessary. More importantly, as the reviewer implies, this maternal knockout allowed us to distinguish whether the T3 action occurs in the mother or the fetus, as we can compare mothers and offspring with intact or defective TRb signalling in the same experiment. To better understand this, we have now labelled the offspring genotype more clearly in Fig 4/5 above the respective bars. Moreover, we have included a better explanation for the selection of this animal model in the results section of the manuscript (p7), which now reads: “To test for a possible role of TH receptor β in the maternal or the offspring compartment, we used TR β ^{-/-} females as dams and compared the offspring to those of TR β ^{+/-} females. Both females were mated with TR β ^{+/-} males, allowing us to analyze the resulting TR β ^{+/-} (hereafter referred to as “controls”, as they are indistinguishable from TR β ^{+/+} wildtypes) and TR β ^{-/-} offspring (Fig 4A).” and the discussion (p14): “Most importantly, despite relying on different mechanisms, the opposite phenotypes in the two groups demonstrate that i) maternal TR β is required for the fetal programming effect of T3 and ii) that the T3 action occurs largely in the mother rather than the fetus.”

3. It sounds that thyroid hormones can pass through the placenta in the introduction section. It is logical to study the direct effects of increased T3 in fetal development. It is odd to know that there was no change in fetal blood T3 and TSH.

Answer: The reviewer is correct that thyroid hormones can pass through the placenta and will increase T3 in the fetal compartment. We did however not provide any fetal T3 or TSH data, due to the fact that the amount of serum in a single embryo is too small to assess T3/T4. However, to test whether the T3 treatment indeed reaches the fetal compartment, we have now collected embryos from T3 treated dams and controls at GD17 and tested the expression of T3 sensitive hepatic genes. The data

revealed distinct changes in gene expression in the embryos of T3 treated dams as compared to controls, demonstrating that the maternal T3 reaches the embryonal liver. The data have now been included in Suppl Fig 1C.

4. What are the effect of T3 treatment and TRb deletion on maternal metabolism? This should be the key information of this ms.

Answer: We appreciate this comment of the reviewer and have now studied maternal metabolism in greater detail. To this end, we have included a new cohort of pregnant T3 treated dams and controls. The data revealed increased body weight gain and food/water intake, as well as increased glucose clearance as expected in the T3 treated pregnant females (Fig 1C+D, below). Moreover, liver and heart

Figure 1

weight were increased in those animals, and the liver displayed the expected upregulation of T3 target genes *Dio1* and *Spot14* (Suppl Fig S1A+B). The effects of T3 in pregnant females are therefore very similar to those observed in normal males (see e.g. Johann et al. 2019 Cell Reports). Finally, we have performed metabolomics studies in the T3 treated females, which revealed 19 altered circulating metabolites (now in Fig 6).

5. Some of the discussion is confusing. For example, RTHb was discussed, but there is no RTHb data in the result section. Most of the conclusions in this section are speculative, lacking data support.

Answer: We apologize for the confusion, which was the result of not properly introducing the abbreviation RTHb. RTHb is “resistance to thyroid hormone beta”, which is the clinical phenotype of humans lacking functional TRb. It is therefore the human equivalent to the TRb knockout mouse model. To avoid further confusion, we have now properly introduced this term in the results on p8, but use the term now exclusively for the human condition to allow a better distinction to the TRb knockout animal model. Furthermore, following the suggestion of the reviewer, we have removed large parts of the discussion on human RTHb with the exception of the HPT axis regulation and possible neuropsychological issues (which was a request of reviewer #3).

Reviewer #3 (Remarks to the Author):

This manuscript reports on the maternal effects of pharmacological and genetic manipulations in the endocrine thyroid hormone system on the adult offspring. The main results are that T3 treatment of pregnant dams leads to hyperactivity of BAT in the adult offspring, and genetic ablation of TRbeta in dams leads to hypoactivity of BAT in adult offspring. These results are potentially of interest to better understand how maternal thyroid status programs offspring metabolism and energy homeostasis. As highlighted in the introduction, in humans maternal effects of hypo- and hyperthyroidism on offspring phenotypes are well known, but the underlying mechanism remain unclear. The experimental design chosen in the present study enables to dissect whether maternal hyperthyroidism depends on the presence of TRbeta in the dam or in the pups.

Although this work is of interest, however, the experimental evidence from mouse models does not go beyond describing an interesting phenomenon, but falls short in delivering mechanisms.

Major points of criticism

The main conclusion is that BAT hyperactivity in offspring is mediated by T3, dependent on TRbeta expression in the dams. The authors therefore postulate an indirect mechanism, involving an unknown circulating factor derived from a TRbeta dependent tissue, like liver, that crosses the placental barrier and conveys programming of BAT. No attempts to identify the unknown factor were undertaken.

Answer: We agree with the reviewer that the identification of the unknown circulating factor is the next logical step. We have therefore conducted a new experiment, in which the pregnant females were killed at gestational day 17, and studied the serum using mass spectrometry based metabolomics. We have identified several circulating factors that were altered by maternal T3 treatment (see Suppl Table 3). For most of those, no information for their role in pregnancy is available in the literature; however, we also found an increase in choline in the T3 treated mothers. Given that the importance of maternal choline for neuronal development is well established, and that the metabolite has been implicated previously in the fetal programming of the brain, including the central control of energy homeostasis, this candidate constitutes a very interesting link that could potentially explain the T3/TRb mediated effect on the fetus. However, validating the regulation of this and the other identified factors by maternal T3 and testing their effect on offspring iBAT function in the context of T3/TRb action would require several years of work and seems beyond the scope of the current manuscript. We have now added the metabolomics analysis to the revised manuscript as Fig 6 and discussed choline as the most relevant candidate.

The present report consolidates the already known involvement of maternal TRbeta in programming of offspring. Maternal programming effects have been reported for humans lacking functional TRbeta (RTHbeta), as referenced by the authors in the introduction (17,18).

Answer: The reviewer is correct that some effects caused by maternal RTHb have already been reported; however, those are primarily in the context of their elevated circulating thyroid hormone levels, which affect the offsprings by altering e.g. the HPT axis. In order to differentiate whether the fetal programming effect is indeed mediated by maternal TRb or rather by the elevated T3 acting on the embryo, a parallel comparison to a T3 treated model as in our study is required. Therefore, our study remains novel in the sense that i) we demonstrate a role of maternal TRb for the programming effect as opposed to the elevated T3 affecting the fetus, and ii) the fetal programming of iBAT thermogenesis has not been shown previously. The latter is of particular interest, given that functional BAT is only found in some humans; however, it remains enigmatic why these individuals possess BAT and others do not. Our study therefore adds a novel aspect to this conundrum.

The introduction is unbalanced as it almost exclusively addresses the negative effects of maternal hypo- and hyperthyroidism on fetal development and the health status of the offspring in humans. Regarding the sole focus of this paper on experimental work in mice and BAT, the introduction fails to introduce important facts on the role of thyroid hormones for BAT physiology, and what we know about hypo- and hyperthyroidism in rodents in respect to maternal effects. Most of the human relevance part should move to the discussion.

Answer: We apologize for the unbalanced introduction and have shortened the human developmental aspect and instead included an introduction on BAT physiology. The human relevance in the discussion has also been shortened in response to the comment of another reviewer.

In addition, there is a major discrepancy between the observed programming effect on BAT in mice and the maternal effects of altered thyroid hormone activity in humans on neuronal / psychiatric phenotypes which is not sufficiently addressed by the authors.

Answer: We agree with the reviewer that we would have expected to see an altered regulation of the HPT axis as well as attention-deficit-hyperactivity disorder, which have been associated with maternal hyperthyroidism in humans (Andersen et al. 2014 BJOG). However, none of these were observed in our mice, as we have normal HPT axis settings and no change in locomotion. With regard to the HPT axis, the observed effects in RTHb human offspring, namely lower TSH in infancy, disappear in adulthood, suggesting that the HPT axis corrects itself over time. As our animals were also studied as adults, the observed normal T3, T4 and Tshb levels are not at odds with the human phenotype. With regard to the psychiatric phenotype, the large-scale population study with >800.000 children referenced above found the risk of ADHD upon maternal hyperthyroidism to be increased by only 18%, which despite being significant, suggests not a straight causal relationship. A correlation between maternal hyperthyroidism and other psychiatric issues such as autism-spectrum disorders has not been observed in this study. As suggested by the reviewer, we have elaborated on these two possible discrepancies between human and mouse phenotype and their possible reasons in the discussion on page 12/13, which now reads: "It is well established that maternal and offspring TSH or fT4 are positively correlated in humans (12), therefore one would have expected lower TSH levels in the offspring of hyperthyroid mothers. This is in fact observed in unaffected infants of mothers with RTHb (33, 34). However, contrary to humans, mice born to TRb knockout mothers showed 2-fold elevated serum TSH in the presence of normal T3 and T4 (30, 35), suggesting an interspecies difference, presumably caused by the difference in relative pregnancy duration between mice and humans. In our

study, we did not observe an effect of maternal hyperthyroidism or TRb knockout on offspring serum T3, T4 or pituitary Tshb mRNA expression. Likewise, markers of peripheral thyroid hormone action such as hepatic Dio1 were unaffected, corresponding with previous studies showing normal tissue T3 and T4 content in these animals (36). Adult humans born to RTHb mothers also display normal baseline serum TSH, T3 and T4 (30, 34), suggesting that any alterations in the hypothalamus-pituitary-thyroid (HPT) axis in early life largely normalizes in adulthood. Interestingly, reduced sensitivity of TSH to thyroid hormone after stimulation seems to persist in the human RTHb offspring – an effect which seems to be transmitted epigenetically along the male line (34). This may be mediated by Dio3 DNA methylation – however, we did not find any effect of maternal thyroid hormone or lack of TRb on Dio3 mRNA expression in pituitary (not shown) or iBAT (not expressed) of our animals, suggesting that in our paradigm the offspring’s HPT axis is not affected. With regard to behavior, large scale population studies reported a significant correlation between maternal hyperthyroidism to attention-deficit-hyperactivity syndrome, but not autism-spectrum disorders (37). However, the adjusted hazard ratio was rather small, suggesting no clear causal relationship.”

Negative data certainly are of indispensable value. However, in this manuscript the reader is exposed to a large amount of mostly negative data. It remains unclear why it is important to include all these negative findings in the main body of the manuscript, and not move them to the supplement.

Answer: We apologize for the large amount of negative data, which were originally included with the intention to provide the most complete picture possible. From the comment we have realized that this may have been too much and possibly distracting from the interesting findings. Thus we have now moved most of the negative data to the supplements or removed them entirely, if they were not of immediate relevance to the study (e.g. heart rate and blood pressure). Since we have added many new results, including the BAT microarray data, the cold exposure and the data on the maternal metabolism, the number of figures remained however the same.

How was the measurement of BAT temperature validated? Was the interscapular area shaved prior to the measurement? Was a reference temperature defined at a more posterior position of the mouse?

Answer: We apologize for not providing sufficient information on the precise infrared analysis of the iBAT. We did not shave the animals, as this can lead to skin inflammation and irritation, which confound the results. Instead, we used Vaseline to clump the hair above the iBAT, which allows a straight line of sight to the skin surface and the BAT temperature becomes independent of the mouse’s position (see figure). The details of this protocol and its validation/comparison to previous methods using UCP1 knockouts and norepinephrine stimulation tests have in the meantime been published (Oelkrug & Mittag, 2021 Scientific Reports).

We have referenced this paper now in the methods section. We did not define a reference temperature on the back, but instead use rectal temperature for normalization, which is much more stable than a back temperature reference position.

figure). The details of this protocol and its validation/comparison to previous methods using UCP1 knockouts and norepinephrine stimulation tests have in the meantime been published (Oelkrug & Mittag, 2021 Scientific Reports).

On page 6 of results, after summarizing the T3 treatment findings, they do not explain properly why they chose to study a possible contribution of maternal TRbeta. The potential mechanistic link between the opposing actions of T3 treatment and genetic ablation of TRbeta in dams on BAT activity in adult offspring is missing. T3 treatment of dams increased BAT temperature (figure 3C) whereas offspring born from TRbeta KO dams showed decreased BAT temperature (figure 6C). However, during fetal development the latter were exposed to elevated maternal T3 levels, alike T3 treatment. On page 6 we read that endogenously hyperthyroid TR-beta KO females were mated, but thyroid hormones were not measured in TR-beta KO dams. Is the T3 elevation comparable to the T3 treatment? Why was there differential response to a similar endocrine stimulus? This discrepancy should be subject to the discussion.

Answer: We thank the reviewer for this comment and the suggestion. We have now measured serum total T3, T4 and pituitary Tshb mRNA in the pregnant dams of all groups to allow for direct comparison. While the TRb KO animals displayed the expected Tshb elevation, a classic hallmark of thyroid hormone resistance RTHb, the levels of T3 and T4 were surprisingly normal (in contrast to non-pregnant females, that still had elevated thyroid hormone, not shown). The data have been included in Fig 1 and Fig 4.

Moreover, we have elaborated in greater detail on the use of TRb KO females in the context of this study. The TRb knockout mothers have been used to test whether maternal TRb would be required for the observed phenotype: if removal of this receptor in the mother would reverse the phenotype (as it did in terms of BAT thermogenesis), we can conclude that it is necessary and that activation of TRb by T3 causes BAT hyperactivity while inhibition of TRb signalling causes BAT hypoactivity. More importantly, as the reviewer suggests, this maternal knockout allowed us to distinguish whether the T3 action occurs in the mother or the fetus, as we can compare offspring and mothers with intact or defective TRb signalling in the same experiment. Thus, we have included a better explanation for the selection of this animal model in the results section of the manuscript (p7), which now reads: "To test for a possible role of TH receptor β in the maternal or the offspring compartment, we used TR β ^{-/-} females as dams and compared the offspring to those of TR β ^{+/-} females. Both females were mated with TR β ^{+/-} males, allowing us to analyze the resulting TR β ^{+/-} (hereafter referred to as "controls", as they are indistinguishable from TR β ^{+/+} wildtypes) and TR β ^{-/-} offspring (Fig 4A)." and the discussion (p14): "Most importantly, despite relying on different mechanisms, the opposite phenotypes in the two groups demonstrate that i) maternal TR β is required for the fetal programming effect of T3 and ii) that the T3 action occurs largely in the mother rather than the fetus."

Alike BAT temperature, BAT cAMP levels were reduced in offspring from TRbeta KO dams. This may be due to reduced sympathetic tone as concluded by the authors, but could also be due to altered expression of adrenergic receptors, G-proteins or adenylylcyclase upstream of cAMP. Notably, cAMP in BAT was not affected by T3-treatment.

Answer: We agree with the reviewer on both aspects. To test whether the cAMP levels could be caused by altered intracellular changes in iBAT such as adrenergic receptors, we performed a microarray based gene expression analysis, which is now included in Fig 5. Surprisingly, only few genes were affected in the offspring of TRb knockout mothers (Fig 5B), and none of which were involved in the propagation

of sympathetic signalling (HeatMap GO 0071875, Suppl Fig 3B, see left figure). With regard to the T3 treated offspring, there were likewise only minor alterations on the mRNA expression landscape in iBAT; however, we found a lower tail temperature accompanying the BAT hyperactivity (now in Fig 5G), resembling the situation found in cold exposure. We therefore continued to challenge the offspring of T3 mothers with 10°C exposure to test whether they would be better prepared to handle the cold, which was indeed the case as there was reduced strain on the other thermogenic tissues muscle and iWAT (Fig 5L). Although we cannot

provide the precise molecular target mediating the higher BAT thermogenesis, the fact that the tail is also more constricted suggests a centrally coordinated alteration in how body temperature is maintained.

On page 9, we read that RMR at thermoneutrality was higher, but this is not supported by the stats (Figure 5E, Supplemental Table), and is in contrast to the description of no difference in RMR on page.

Answer: The reviewer is mistaken, there was no reference to RMR at thermoneutrality on page 9 of the original manuscript, we only stated on page 8 that “There was also no difference in the metabolic rate at rest in 23°C or 30°C (Fig 5E).” and that is in line with the data shown in the old figure 5E. In the revised version of the manuscript, we only provide RMR at 30°C in Fig 3E.

Regarding the energy balance data, duration of measurements for food intake are too short to detect differences between treatment groups, as food intake data normally show more day-by-day variation than energy expenditure recordings.

Answer: The food intake data, which were included in the manuscript, had been taken in parallel to the energy expenditure in the TSE system to allow for better comparison between uptake and expenditure. We completely agree that more subtle changes in food intake can only be properly quantified if the accumulated food intake is recorded over several weeks. We therefore recorded cumulative food intake in a new cohort of mT3 and control offspring starting at week 5 until week 10. The results corroborated the TSE recordings by showing no difference between the groups ($p=0.82$ for mT3, RM-ANOVA). Given that there is of course an effect of age due to the increasing body weight, we would prefer to keep the data gathered by the TSE, as they match the energy expenditure.

The presentation of metabolic data is not well structured. It remains unclear whether there are significant maternal effects on metabolic rates of adult offspring, as the applied analyses of metabolic data are not state of the art. First, oxygen consumption and DEE data are presented per mouse (figures 2C-F and 5C,D). These should be directly compared with the respective food & water intake data (figure 2I, 5F), and with the locomotor activity data. Energy in and energy out should be expressed as energy units (kcal or kJ). While DEE, food & water consumption are expressed per mouse, why did the authors choose to show mass-specific ratios for RMR (figures 2G,H, and 5E). Several reviews address the problems of data analyses, and the use of mass-specific ratios has been heavily criticized in the past. One solution could be to calculate ratios per gram of lean mass. However, ANCOVA or GLM based stats are recommended. One solution would be to use the CalR platform for state of the art data analyses.

Answer: We appreciate this expert comment of the reviewer. In fact, we had performed all respective ANCOVA analyses (similar to our previous publications Oelkrug et al. 2020 Cell Reports), but we opted to not include those graphs as they did not reveal any additional information, as there was no difference in body weight or lean mass in the groups that could skew the interpretation.

To take this concern into account, we have now changed the food intake and the RMR at 30°C to energy units (kJ) so that energy in (Fig 3F and 4J) and energy out (Fig 3B+D+E and 4I) can be compared. Moreover, in the main figure all data are now consistently presented as per mouse, while the supplements show the ANCOVA data using body mass as covariate (Suppl Fig S1F and S2C) and the normalization to lean body mass (Suppl Fig S1F and S2C). The respective statistical details have been added to Suppl Table 1. Both types of analyses did not show any significant effect of maternal T3 treatment.

Why do body composition percentages not add up to 100%? Small differences in body composition may be detectable / verifiable by using proper statistical methods (ANCOVA) instead of mass-specific ratios (percentages in figures 1E and 4D,E).

Answer: The difference to 100% consist of fur, bones, etc., which are not included in lean, fat or free fluid. With regard to small differences, we agree with the reviewer. However, instead of using statistical normalization, we routinely measure the individual weight of the different tissues when we sacrifice the animals, as in contrast to NMR, we can then also differentiate between the different fat pads. This is of particular relevance in the context of thermoregulation, as we can find minor details in e.g. the thermogenically active inguinal adipose tissue (iWAT) that differs functionally from the non-thermogenic eWAT depots (see Fig 5L) – which could not be detected by the NMR.

Did T3 treatment or the genotype of dams have an effect on duration of gestation. Did these interventions delay or accelerate the timing of parturition?

Answer: We thank the reviewer for this comment and repeated the experiment to obtain data on the gestation time in our models. Indeed, the T3 treated dams gave birth one day later as compared to controls (GD20 vs GD19), which likely contributes to the slightly higher birthweight in these offspring (that difference however disappears later in life). The information has been added to the manuscript on p6.

REVIEWER COMMENTS

Reviewer #3 (Remarks to the Author):

Nice piece of work! All my points of criticism were addressed by your thorough revision.

Reviewer #4 (Remarks to the Author):

The authors have performed an extensive set of additional experiments, have studied the effect of a lack of maternal TRB in the offspring in much more depth and have performed a number of additional studies in search of a mechanism to explain their results. Overall the paper has improved significantly. I have a number of minor comments that need to be addressed before the manuscript is ready for publication.

- a number of gene names are incorrect, for example the NCBI gene ID for spot14 is Thrsp, and for malic enzyme is Me1. Please adjust this throughout the manuscript, figures and supplemental data and double check your other gene names.

- Unaffected infants of RTHB mothers in fact have central hypothyroidism due to an altered HPT axis point (see PMID: 28586435). Your discussion suggests they have a normal HPT axis, please adjust.

- The manuscript occasionally refers to the TRBKO mice as RTH beta animals, as this is a different mechanism (a dominant negative receptor that cannot bind T3 vs the lack of a receptor) this is not the correct term, please consequently refer to your mice as TRB knockouts, or mice lacking TRB etc.

- Although the discussion reads well and has improved from the last version, it is unnecessarily long and expands on certain rather general subjects too much, almost more like a review than a discussion in some parts. Given its length please shorten some of the sections and discuss only what is directly relevant for your results. For example the section on TH metabolism and behaviour as you do not test behaviour in your model, so this discussion has no place in this paper, and could be much more concise in your description of the TH levels found in your experiment.

Reviewer #5 (Remarks to the Author):

Oelkrug et al present data examining the activation of maternal thyroid hormone receptor beta and its impact on offspring brown fat thermogenesis.

How bioavailable is the T3 provided in drinking water?

The authors should provide more details in relation to the early life situation. Was there a difference in M/F ratio, pregnancy loss, number of pups per litter (were pups culled back to a certain number at P1 or P2), etc these factors are all extremely important in terms of accurately interpreting the adult offspring results.

What were the n-numbers for these experiments considering that the maternal exposure is the statistical unit. It seems as though the offspring are considered as distinct statistical units which is not appropriate for this type of study design. Were power calculations carried out to determine the

number of animals used?

GD17 are fetuses rather than embryos.

Why were IP rather than oral GTT carried out? Studies have clearly demonstrated that the oral route is much more physiological than IP.

Just as a super pedantic point, gender is a social construct rather than a biological term, sex is the correct term to use.

It seems hugely shortsighted to focus on one sex above the other, especially while interesting sex specific effects were observed in the mT3 part of the study. The rationale for doing this is weak and omission of females in this part of the study massively weakens the impact of the results.

Lines 271-275 the authors mention links between adhd and asd, surely this experiment would have been a nice avenue to explore this further. Rather than the maternal care experiments (which seem a bit out of place in the general focus of this work) there are some easy experiments such as measurements of ultrasonic vocalisations in newborn pups etc that would potentially tease this question out a bit further.

If you're looking for clues in relation to altered glucose handling perhaps the white adipose tissue would be more appropriate than the BAT. Did the authors examine the potential of 'browning' of white adipose tissue?

Reviewer #3 (Remarks to the Author):

Nice piece of work! All my points of criticism were addressed by your thorough revision.

A: We thank the reviewer for the time and the encouraging response!

Reviewer #4 (Remarks to the Author):

The authors have performed an extensive set of additional experiments, have studied the effect of a lack of maternal TRB in the offspring in much more depth and have performed a number of additional studies in search of a mechanism to explain their results. Overall the paper has improved significantly. I have a number of minor comments that need to be addressed before the manuscript is ready for publication.

- a number of gene names are incorrect, for example the NCBI gene ID for spot14 is Thrsp, and for malic enzyme is Me1. Please adjust this throughout the manuscript, figures and supplemental data and double check your other gene names.

A: We thank the reviewer for this remark and have corrected all gene names accordingly.

- Unaffected infants of RTHB mothers in fact have central hypothyroidism due to an altered HPT axis point (see PMID: 28586435). Your discussion suggests they have a normal HPT axis, please adjust.

A: The reviewer is correct in pointing this out. As can be seen from Table 3 in the indicated reference, T4, T3 and TSH levels are not significantly different in unaffected offspring of RTH β mothers as compared to those of RTH β fathers, which was what we were referring to in the discussion. However, the response to T3 seems to be blunted, indicating a central resistance that suggests a defective response in the HPT axis despite the normal TH levels under unchallenged conditions. We have therefore removed the incorrect part of the sentence and now state: "in adulthood, offspring of RTH β mothers also display normal baseline serum TSH, T3 and T4 (30, 34), but a reduced sensitivity of TSH to thyroid hormone – an effect which seems to be transmitted epigenetically along the male line (34)."

- The manuscript occasionally refers to the TRBKO mice as RTH beta animals, as this is a different mechanism (a dominant negative receptor that cannot bind T3 vs the lack of a receptor) this is not the correct term, please consequently refer to your mice as TRB knockouts, or mice lacking TRB etc.

A: The original case from 1967 that coined the term "resistance to thyroid hormone" was actually a family with a homozygous KO of the THR β gene (Refetoff et al. 1967). Thus, technically the term resistance to thyroid hormone beta includes both the mutation as well as the deletion (see Moran et al. 2022 Clin Endoc, <https://doi.org/10.1111/cen.14817>). However, we understand the point of the reviewer that this distinction could be helpful in better understanding the consequences and we have changed the text accordingly on p13 to "offspring of TR β knockout mothers".

- Although the discussion reads well and has improved from the last version, it is unnecessarily long and expands on certain rather general subjects too much, almost more like a review than a discussion in some parts. Given its length please shorten some of the sections and discuss only what is directly relevant for your results. For example the section on TH metabolism and behaviour as you do not test behaviour in your model, so this discussion has no place in this paper, and could be much more concise in your description of the TH levels found in your experiment.

A: We agree with reviewer and have shortened the discussion in general, and specifically eliminated the section discussing the effects on behavior.

Reviewer #5 (Remarks to the Author):

Oelkrug et al present data examining the activation of maternal thyroid hormone receptor beta and its impact on offspring brown fat thermogenesis.

How bioavailable is the T3 provided in drinking water?

A: Both forms of thyroid hormone, T4 and T3, exhibit generally good bioavailability in oral applications, which is also the standard therapy form for humans as tablets of Levothyroxin (T4) or Liotrix (T3 and T4 combination). In mice, T3 treatment in drinking water is preferred over injection, as the short half-life of T3 would require at least twice daily injections and thus results in rapid supraphysiological peaks in systemic T3 (see Guidelines of the American Thyroid Association to Investigate Thyroid Hormone Economy in Rodents, Bianco et al. 2014 Thyroid). Repeated injections and high peak T3 levels are especially undesirable in pregnant females, as they risk the termination of pregnancy, which is why we opted for T3 in drinking water. The efficiency of the treatment was demonstrated by the 6-fold elevation of serum T3 and a complete suppression of pituitary TSH (Fig 1b), which underlines the feasibility of the approach and the bioavailability in drinking water.

The authors should provide more details in relation to the early life situation. Was there a difference in M/F ratio, pregnancy loss, number of pups per litter (were pups culled back to a certain number at P1 or P2), etc these factors are all extremely important in terms of accurately interpreting the adult offspring results.

A: The reviewer is correct that this information is crucial for assessing our results, and we apologize for not including sufficient detail in our manuscript. Regarding the M/F ratio, there was no significant differences in T3-treated versus control mothers, but a trend towards more male pups in TR β knockout mothers. These data have been included now as Suppl Fig 1d and Suppl Fig 2a.

There was also no significant difference in litter size of T3 treated mothers or TR β knockout mothers. These data were included in the original manuscript Fig 1f for the T3 treated mothers, and the TR β knockout data have now been added as Suppl Fig 2a. With regard to the numbers per pups per litter, they were reduced to a maximum of 8 pups per group at P1 or 2 if necessary. In addition, mothers with fewer than 6 pups per litter were excluded from further analysis to avoid differences in maternal care. We apologize for omitting this information, which has now been added to the Material and Methods section.

What were the n-numbers for these experiments considering that the maternal exposure is the statistical unit. It seems as though the offspring are considered as distinct statistical units which is not appropriate for this type of study design. Were power calculations carried out to determine the number of animals used?

A: We agree with the reviewer; however, this issue is currently controversially discussed not only in the context of animal welfare (discarding all animals except one per litter after weaning is not accepted by our current legislation), but also with regard to a possible bias that is introduced by selecting a random single animal per litter. We therefore opted for an intermediate approach by taking several animals per litter but at the same time ensuring that the final number of animals per experiment is derived from several litters, which is similar to what has been done in several other publications using a maternal intervention paradigm in Nature Communications:

<https://www.nature.com/articles/s41467-022-32230-2>

<https://www.nature.com/articles/s41593-020-00762-9>

<https://www.nature.com/articles/s41467-021-24719-z>

<https://www.nature.com/articles/s41467-021-25220-3>

Power calculations are required by law for all animal ethical applications in Germany, and were conducted with the above experimental design in mind and approved by the legal authorities, as given in the method sections. Nevertheless, we agree with the reviewer that more details on the number of litters should be provided. We therefore included additional information to each figure legend stating the number of biological replicates and the number of litters used to generate these replicates, similar to what is presented in the other Nature Communications publications referenced above.

GD17 are fetuses rather than embryos.

A: This is an interesting issue raised by the reviewer, and we agree that the terminology is less clear in mice than in humans. Mice are in contrast to humans born prematurely, therefore the developmental transition from embryo to fetus is not as distinct. Consequently, most developmental researchers have opted to use the term embryo until birth (Crawford et al. 2010 Toxicol Pathol and Kaufmann 1999

“Atlas of Mouse Development”). Therefore, to avoid unnecessary confusion, we have now corrected the manuscript and use the term “embryo” consistently for the prenatal mouse and the term “fetus” only when referring to the human condition.

Why were IP rather than oral GTT carried out? Studies have clearly demonstrated that the oral route is much more physiological than IP.

A: The reviewer is correct that the oGTT is more physiological and includes the incretin response, which is missed with the ipGTT (Small et al. 2022 Mol Met). However, as we were not interested in the incretin response and in addition tested insulin sensitivity separately, we opted for the ipGTT, as it is easier to perform and also recommended by the International Mouse Phenotyping Consortium (www.mousephenotype.org). We therefore adopted an ipGTT protocol that has also been used successfully in several other high impact publications (e.g. Quarta et al. 2022 Nat Met).

Just as a super pedantic point, gender is a social construct rather than a biological term, sex is the correct term to use.

A: We appreciate the comment of the reviewer and have replaced all “gender” by “sex”.

It seems hugely shortsighted to focus on one sex above the other, especially while interesting sex specific effects were observed in the mT3 part of the study. The rationale for doing this is weak and omission of females in this part of the study massively weakens the impact of the results.

A: We agree with the reviewer that ideally the second part of the study should have also included females. However, since this experimental paradigm was already quite complex, involving mothers and offspring of two different genotypes, we decided to focus our analysis on males only, as they exhibited both phenotypic features of maternal T3 treatment, namely altered BAT thermogenesis and altered glucose tolerance, while in females glucose metabolism was apparently not affected. This sexual dimorphism is not unusual for studies addressing fetal programming effects, as males are often more affected than females (reviewed in e.g. Dunn et al. 2011 Horm & Behavior; Miranda & Sousa 2018 Brain & Behavior). Consequently, based on the literature and our results of the T3 treated mothers, we expected the more pronounced phenotype in the males. Unfortunately, we did not take any data on TR β offspring females and we will not be able to repeat the entire study in females in the timeframe of 4 weeks given for the revision. This will instead be the focus of future studies. To take the important point of the reviewer into account, we have acknowledged this limitation in the discussion, which now reads (p16): “However, additional studies in female offspring will be required to characterize further the sexual dimorphism of the effect.”

Lines 271-275 the authors mention links between adhd and asd, surely this experiment would have been a nice avenue to explore this further. Rather than the maternal care experiments (which seem a bit out of place in the general focus of this work) there are some easy experiments such as measurements of ultrasonic vocalisations in newborn pups etc that would potentially tease this question out a bit further.

A: We agree with the reviewer that additional experiments on a possible ADHD/ASD disorder in the TR β knockout offspring would be extremely interesting to conduct, and ultrasound vocalization would be a very valuable approach to this question. Unfortunately, we did not perform those experiments in the pups. We did however measure overall activity in these offspring, which was not significant between the groups, indicating that at least under our experimental conditions, no hyperactivity is induced in the offspring of hyperthyroid TR β knockout mothers. Given that reviewer #4 felt that the behavioral parts were outside the scope of the manuscript and the discussion, we added the activity data to the Supplementary Figure 2, but did not elaborate further on the issue in the discussion.

If you're looking for clues in relation to altered glucose handling perhaps the white adipose tissue would be more appropriate than the BAT. Did the authors examine the potential of 'browning' of white adipose tissue?

A: This is an excellent suggestion of the reviewer, as it could indeed explain the observed phenotype! We have thus now analyzed gene expression in iWAT for markers of browning in the offspring of T3 treated mothers. Unfortunately, however, the results by qPCR yield levels of Ucp1 mRNA close to background in either condition (controls average ct $35,8 \pm 2,0$ versus T3 offspring $34,6 \pm 3,3$, compared to ct values of around 22 in properly browned iWAT (Johann et al. 2019 Cell Reports) or 18-19 in iBAT). Furthermore, no UCP1 protein could be detected in either condition by Western Blot. Taken together, this suggests that iWAT is not browned and thus not a possible source of the altered glucose handling in the T3 offspring males. We have added this information to the results, which now reads: "The altered glucose metabolism was not caused by browning of inguinal white adipose tissue, as we could not detect any UCP1 protein in this tissue and Ucp1 mRNA levels were at background levels in either condition (data not shown)."